# ∇-REASONER: LLM REASONING VIA TEST-TIME GRADIENT DESCENT IN LATENT SPACE

**Peihao Wang**[1*], **Ruisi Cai**[1*], **Zhen Wang**[2], **Hongyuan Mei**[3], **Qiang Liu**[1],
**Pan Li**[4], **Zhangyang Wang**[1]
[1]The University of Texas at Austin, [2]UC San Diego, [3]TTIC, [4]Georgia Tech
{peihaowang,ruisi.cai,lqiang,atlaswang}@utexas.edu, zhw085@ucsd.edu
hongyuan@ttic.edu, panli@gatech.edu

⊙ https://github.com/VITA-Group/Nabla-Reasoner

## ABSTRACT

Scaling inference-time compute for Large Language Models (LLMs) has unlocked unprecedented reasoning capabilities. However, existing inference-time scaling methods typically rely on inefficient and suboptimal discrete search algorithms or trial-and-error prompting to improve the online policy. In this paper, we propose ∇-Reasoner, an iterative generation framework that integrates differentiable optimization over token logits into the decoding loop to refine the policy on the fly. Our core component, Differentiable Textual Optimization (DTO), leverages gradient signals from both the LLM's likelihood and a reward model to refine textual representations. ∇-Reasoner further incorporates rejection sampling and acceleration design to robustify and speed up decoding. Theoretically, we show that performing inference-time gradient descent in the sample space to maximize reward is dual to aligning an LLM policy via KL-regularized reinforcement learning. Empirically, ∇-Reasoner achieves over 20% accuracy improvement on a challenging mathematical reasoning benchmark, while reducing number of model calls by approximately 10-40% compared to strong baselines. Overall, our work introduces a paradigm shift from zeroth-order search to first-order optimization at test time, offering a cost-effective path to amplify LLM reasoning.

## 1 INTRODUCTION

Large Language Models (LLMs) have unlocked remarkable reasoning capabilities (Radford et al., 2018; 2019; Brown et al., 2020), enabling machines to tackle challenges considered exclusive to human cognition, such as solving complex mathematical problems (Cobbe et al., 2021; Lewkowycz et al., 2022; Uesato et al., 2022; Lee et al., 2023; Yang et al., 2024b) and executing long-horizon planning (Liu et al., 2023a; Valmeekam et al., 2023; Song et al., 2023). Such capabilities arise through large-scale pre-training on massive datasets, followed by careful post-training alignment (Wei et al., 2022a; Ouyang et al., 2022; Guo et al., 2025). A prevailing observation has indicated that scaling both model size and training data leads to continual improvements in LLM reasoning ability (Kaplan et al., 2020; Hoffmann et al., 2022).

Nevertheless, recent empirical findings increasingly suggest that scaling inference-time computation can be also crucial and perhaps more cost-effective than expanding pretraining to further enhance reasoning and problem-solving abilities (Snell et al., 2024). Chain-of-Thought (CoT) (Wei et al., 2022b) demonstrates that prompting LLMs at the test time to generate longer sequences with intermediate reasoning steps significantly improves their reasoning accuracy. Built on CoT, Wang et al. (2022) further scales the inference compute by sampling multiple reasoning chains and selecting the most consistent one, leading to enhanced performance. More recently, inference-time scaling has been augmented with reward models to refine reasoning quality. Notable approaches such as Tree-of-Thought (ToT) (Yao et al., 2024) and Reasoning-as-Planning (RAP) (Hao et al., 2023) cast LLM reasoning as a decision-making problem and employ strategic sampling algorithms to estimate the reward-to-go, thereby refining the sequential prediction policy at each decoding step. Underlying these approaches are extensive prompting-based search procedures that traverse the sequence space,

---
*Equal contribution.

with the LLM serving as a guiding heuristic. However, such approaches often struggle to adequately explore the sample space and thus become sensitive to sparse and noisy reward signals as reasoning chains grow longer and the search space expands exponentially. Consequently, their performance tends to saturate even when inference-time computation is substantially increased.

While existing methods fall into zeroth-order algorithms that rely solely on reward values, we note that first-order methods, providing directional guidance for optimization, can be even more effective in searching for optimal solutions, overcoming the sparsity of the reward landscape (see an intuitive comparison in Fig. 1). In fact, gradient information is readily available during the LLM reasoning process, as both the LLM and reward function can be differentiable. In this paper, we introduce $\nabla$-Reasoner, a novel reasoning algorithm that applies inference-time gradient descent in the sample space to refine the outputs of a base policy prior to next-token prediction. The overall pipeline follows an iterative decoding process. At each step, the language model first generates a full completion together with its per-token logits, serving as the initial rollout. The core component, termed Differentiable Textual Optimization (DTO), then refines these token logits via gradient descent.

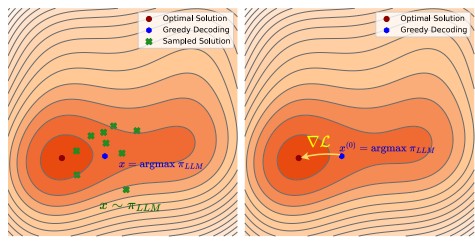

Figure 1: LLM reasoning can be formulated as a maximization problem over the reward landscape. (left) Traditional inference-time scaling methods are zeroth-order methods, sampling numerous candidate responses and evaluating them to identify higher-quality solutions. (right) This paper introduces a first-order method for inference-time scaling, where reward gradients are directly leveraged to guide the search process toward highly rewarding responses.

DTO formulates the reasoning process as a continuous optimization problem over the reward landscape, directly leveraging gradients to refine textual representations. Specifically, DTO applies gradient descent to optimize the initial logit vectors from the base policy under an objective that combines the reward function with the sequence-level log-likelihood estimated by the language model. To enable end-to-end differentiability, we employ the straight-through estimator to map logit parameters into one-hot token vectors (Bengio et al., 2013). In this formulation, the reward function provides directional signals that guide tokens toward high-reward regions, while the log-likelihood term regularizes the tuned sequences to remain fluent and consistent with the pre-trained LLM distribution (Hoang et al., 2017; Qin et al., 2022; Kumar et al., 2022).

After refining the logits with DTO, $\nabla$-Reasoner treats the optimized logits of the first token as an improved policy and samples the next-token prediction from this updated distribution. We further integrate $\nabla$-Reasoner with rejection sampling, which accepts the token drawn from the refined policy only if it can yield continuation with higher reward; otherwise, the method reverts to the initial choice. Through iteratively interleaving decoding and refinement, $\nabla$-Reasoner scales inference-time reasoning by allocating additional computation to improve the LLM policy via gradient-based updates on the output space, efficiently backended by the parallel execution of transformer models. To further increase decoding throughput, we introduce a set of acceleration strategies that selectively skip tokens unlikely to benefit from DTO and reuse rollouts shared among decoding steps.

Theoretically, we show that DTO enables bidirectional gradient propagation along the sequence, facilitating global modifications that are crucial for effective reasoning (Bachmann & Nagarajan, 2024; Hao et al., 2023). Furthermore, we establish a close connection between DTO and RL algorithms (Schulman et al., 2017; Ouyang et al., 2022; Guo et al., 2025). We prove that sampling from an optimized LLM trained with RL is equivalent to directly drawing samples from the reference LLM and subsequently refining them through the gradient flow induced by DTO. This insight provides a new theoretical perspective for test-time approaches for reasoning.

Empirically, $\nabla$-Reasoner significantly enhances the mathematical reasoning capabilities by 10-40% across multiple models and benchmarks. It consistently outperforms strong inference-time baselines such as Best-of-N and RAP (Hao et al., 2023), while achieving accuracy on par with more costly training-based methods (e.g., GRPO). We further show that $\nabla$-Reasoner scales compute more effectively: by leveraging parallelized execution of attention, it can utilize more compute per model forward pass. Henceforth, when comparing with sampling-only methods (e.g., BoN), $\nabla$-Reasoner achieves superior results while reducing the number of model calls by up to 40.2%.

## 2 PRELIMINARIES

In this section, we formulate LLM reasoning as a decision-making problem, introducing the necessary notations and common approaches to address this problem along the way.

**Notations.** Let $\mathcal{V} = \{\boldsymbol{\delta}_i \in \mathbb{R}^{|\mathcal{V}|} : i \in [|\mathcal{V}|]\}$ be the vocabulary set, where $\boldsymbol{\delta}_i$ is the $i$-th canonical basis and $|\mathcal{V}|$ is the vocabulary size. We denote a sequence over this vocabulary as $\boldsymbol{x} = [\boldsymbol{x}_1, \cdots, \boldsymbol{x}_{|\boldsymbol{x}|}]$ where $\boldsymbol{x}_l \in \mathcal{V}$ is the $l$-th token for every $i \in [|\boldsymbol{x}|]$ and $|\boldsymbol{x}|$ represents the length of the sequence. We also denote $\boldsymbol{x}_{\leq i}$ as the subsequence up to and including the $i$-th token, expressed as $[\boldsymbol{x}_1, \cdots, \boldsymbol{x}_i]$. The space of all sequences with finite length is given by $\mathcal{V}^* = \bigcup_{l \in \mathbb{N}} \mathcal{V}^l$.

**Language Models and Reward Models.** In this paper, we focus on autoregressive language models (Radford et al., 2018; 2019; Brown et al., 2020), denoted as $\pi_{LLM} : \mathcal{V}^* \rightarrow [0, 1]$. The language model can characterize the conditional probability of a question-answer pair. Given a pair of questions and answers $\boldsymbol{x}, \boldsymbol{y} \in \mathcal{V}^*$, the model estimates their likelihood by the following factorization: $\pi_{LLM}(\boldsymbol{y}|\boldsymbol{x}) = \prod_{i=1}^{|\boldsymbol{y}|} \pi_{LLM}(\boldsymbol{y}_i|\boldsymbol{y}_{\leq i-1}, \boldsymbol{x})$. We also denote $\text{Cat}\left(\pi_{LLM}(\cdot|\boldsymbol{y}_{\leq i-1}, \boldsymbol{x})\right) \in [0, 1]^{|\mathcal{V}|}$ as the categorical distribution of $\boldsymbol{y}_i$ given the prefix $\boldsymbol{y}_{\leq i-1}$ and $\boldsymbol{x}$. In addition, we define a reward model as the function $r : \mathcal{V}^* \rightarrow \mathbb{R}$. $r(\boldsymbol{y}|\boldsymbol{x})$ evaluates the correctness of the response $\boldsymbol{y}$ for question $\boldsymbol{x}$. In this work, we mainly focus on *outcome reward* (Cobbe et al., 2021), which is often a sequence classifier, offering an overall score for the entire response sequence. Our proposed method can also generalize to process reward (Lightman et al., 2023).

**Reasoning as Decision Making.** Reasoning with LLMs can be framed as a search algorithm that aims to identify a high-rewarding response: $\arg\min_{\boldsymbol{y} \in \mathcal{V}^*} -R(\boldsymbol{y}|\boldsymbol{x})$. Due to the combinatorial nature of this optimization, directly finding the optimal solution is intractable, as the search space grows exponentially with the sequence length, i.e., $|\mathcal{V}|^{|\boldsymbol{y}|}$. In autoregressive decoding, this challenge reduces to making sequential decisions for the next token, each of which must ultimately contribute to minimizing $-R(\boldsymbol{y}|\boldsymbol{x})$. This decision process is often formalized via a Bellman equation:

$$\pi_{LLM}^*(\cdot|\boldsymbol{y}_{\leq i-1}, \boldsymbol{x}) = \arg\max_{\boldsymbol{y}_i \in \mathcal{V}} \mathbb{E}_{\boldsymbol{y}_{\geq i+1} \sim \pi_{LLM}^*(\cdot|\boldsymbol{y}_i, \boldsymbol{y}_{\leq i-1}, \boldsymbol{x})}[r(\boldsymbol{y}_{\leq i-1}, \boldsymbol{y}_i, \boldsymbol{y}_{\geq i+1}|\boldsymbol{x})], \quad (1)$$

where $\pi_{LLM}^*$ is a refined version of the original policy $\pi_{LLM}$. The expected reward-to-go in Eq. 1 is also known as the Q-function. The recursive structure of this formulation implies that greedy decoding is not globally optimal, and identifying the optimal next-token prediction inherently requires look-ahead rollouts and backtracking (Yao et al., 2024; Hao et al., 2023; Besta et al., 2024).

**Existing Approaches.** Current techniques tackling LLM reasoning via decision making can be broadly categorized into training-time and inference-time methods. Training-time approaches include supervised fine-tuning (SFT) as well as model-free policy optimization techniques, such as Schulman et al. (2017); Guo et al. (2025); Rafailov et al. (2024). Our focus is on *inference-time methods*, which aim to improve the decoding process of an LLM without additional training. These methods are typically model-based and value-based, seeking to refine an existing policy by directly solving the Bellman equation. For example, Best-of-N (BoN) (Stiennon et al., 2020) tackles Eq. 1 by sampling $N$ independent full trajectories from a base policy $\boldsymbol{y}^{(1)}, \cdots, \boldsymbol{y}^{(N)} \sim \pi_{LLM}(\cdot|\boldsymbol{x})$, and selecting the one with the highest reward $\boldsymbol{y}^* = \arg\max_{\boldsymbol{y} \in \{\boldsymbol{y}^{(1)}, \cdots, \boldsymbol{y}^{(N)}\}} r(\boldsymbol{y}|\boldsymbol{x})$. More structured approaches, such as Tree-of-Thoughts (ToT) (Yao et al., 2024) and Reasoning-as-Planning (RAP) (Hao et al., 2023), explore the solution space and approximate Q-functions stochastically on the fly through rollouts and recursive evaluation.

## 3 REASONING WITH GRADIENT-DRIVEN DECODING

**Overview.** In this section, we introduce $\nabla$-Reasoner, a novel reasoning algorithm that scales inference-time computation by performing gradient descent in the sample space to refine the outputs of a base policy. The overall pipeline, as illustrated in Fig. 2, is structured as an iterative decoding process. Given a prefix $\boldsymbol{x}$, the model first generates an initial response $\boldsymbol{y}^{(0)}$. $\nabla$-Reasoner then represents the generated sequence through its per-token pre-softmax logits $\boldsymbol{z}^{(0)}$ and optimizes these logits via gradient descent to maximize the sequence-level reward $r(\boldsymbol{y}|\boldsymbol{x})$ (Sec. 3.1). After

**Algorithm 1** $\nabla$-Reasoner: Decoding with DTO

**Require:** Prompt $\boldsymbol{x}$, language model $\pi_{LLM}$, reward model $r$, stop criteria StopCriteria$(\cdot)$.
1: **repeat**
2:    $\boldsymbol{y}, \boldsymbol{z} \sim \pi_{LLM}(\cdot|\boldsymbol{x})$
3:    $\widetilde{\boldsymbol{z}} \leftarrow \mathrm{DTO}(\boldsymbol{x}, \boldsymbol{z}, \pi_{LLM}, r)$.
4:    $\widetilde{\boldsymbol{y}}_1 \sim \mathrm{softmax}(\widetilde{\boldsymbol{z}}_1/\tau)$.
5:    **if** $\widetilde{\boldsymbol{y}}_1 \neq \boldsymbol{y}_1$ **then**
6:      $\widetilde{\boldsymbol{y}}, \widetilde{\boldsymbol{z}} \sim \pi_{LLM}(\cdot|\widetilde{\boldsymbol{y}}_1, \boldsymbol{x})$
7:      **if** $r(\widetilde{\boldsymbol{y}}, \widetilde{\boldsymbol{y}}_1|\boldsymbol{x}) > r(\boldsymbol{y}|\boldsymbol{x})$ **then**
8:        $\boldsymbol{x} \leftarrow \mathrm{concat}[\boldsymbol{x}, \widetilde{\boldsymbol{y}}_1]$
9:      **else**
10:       $\boldsymbol{x} \leftarrow \mathrm{concat}[\boldsymbol{x}, \boldsymbol{y}_1]$
11:      **end if**
12:    **else**
13:      $\boldsymbol{x} \leftarrow \mathrm{concat}[\boldsymbol{x}, \boldsymbol{y}_1]$
14:    **end if**
15: **until** StopCriteria$(\boldsymbol{x})$
16: **return** $\boldsymbol{x}$

**Algorithm 2** Differentiable Textual Optimization (DTO)

**Require:** Prefix $\boldsymbol{x}$, initial logits $\boldsymbol{z}$, language model $\pi_{LLM}$, reward model $r$, and the number of training steps $T$.
1: $\boldsymbol{z}^{(1)} \leftarrow \boldsymbol{z}$
2: **for** $t = 1, \cdots, T$ **do**
3:    **for** every $i = 1, \cdots, |\boldsymbol{y}|$ **do**
4:      $j^* \leftarrow \arg\max_{j \in [|\mathcal{V}|]} \boldsymbol{z}_{ij}^{(t)}$
5:      $\boldsymbol{y}_i^{(t)} \leftarrow \boldsymbol{\delta}_{j^*} + \mathrm{softmax}(\boldsymbol{z}_i^{(t)}/\tau) -$
     $\mathrm{StopGrad}(\mathrm{softmax}(\boldsymbol{z}_i^{(t)}/\tau))$
6:    **end for**
7:    $\mathcal{L}_{nll} = -\sum_i \log \pi_{LLM}(\boldsymbol{y}^{(t)}|\boldsymbol{y}_{\leq i-1}^{(t)}, \boldsymbol{x})$
8:    $\mathcal{L}_{reward} = -r(\boldsymbol{y}^{(t)}|\boldsymbol{x})$.
9:    $\mathcal{L} = \mathcal{L}_{nll} + \lambda\mathcal{L}_{reward}$.
10:   $\boldsymbol{z}^{(t+1)} \leftarrow \boldsymbol{z}^{(t)} - \eta\nabla_{\boldsymbol{z}}\mathcal{L}$.
11: **end for**
12: **return** $\boldsymbol{z}^{(T)}$

Figure 2: Basic implementation of $\nabla$-Reasoner. $\nabla$-Reasoner is an iterative decoding algorithm driven by DTO. At each decoding step, DTO applies gradient descent on the logits initialized from the base model to optimize a reward-informed loss to refine the policy. The updated policy is then combined with rejection sampling, leading to high-reward responses. The pseudocode for the full implementation with acceleration techniques (Sec. 3.3) is deferred to Appendix B.

optimization, $\nabla$-Reasoner resamples the *first token* of the generated sequence using the fine-tuned logits $\widetilde{\boldsymbol{z}}_1$. If the resampled token differs from the original, the subsequent tokens are regenerated, and this candidate token is accepted only if its yielded response achieves a higher reward under $r(\cdot|\boldsymbol{x})$ (Sec. 3.2). The procedure then proceeds to the next token by incorporating the first generated token into the prefix, and repeating this optimization-and-resampling loop. $\nabla$-Reasoner scales inference-time reasoning by allocating additional computation to optimize the policy's outputs via iterative gradient descent. To further improve efficiency, we propose a series of system co-design strategies that selectively skip tokens unlikely to benefit from optimization and reuse model outputs and KV caches to accelerate decoding (Sec. 3.3).

## 3.1 DIFFERENTIABLE TEXTUAL OPTIMIZATION

The core step of our algorithm is leveraging gradient information to refine an initial response generated by the base policy. Existing reward-guided decoding methods (Wei et al., 2022b; Wang et al., 2022; Yao et al., 2024; Hao et al., 2023) can be regarded as zeroth-order approaches, as they rely solely on reward values. However, reward feedback is often sparse, and searching for improved solutions based only on scalar reward values can be sample-inefficient, particularly when the base policy is weak. We note that most reward models are inherently differentiable, as they are typically implemented with transformer-based sequence classifiers (Stiennon et al., 2020; Ouyang et al., 2022; Dong et al., 2024). This opens the door to exploiting not just reward values but also reward gradients, which provide richer directional information to guide samples toward high-reward regions. Motivated by this, we reformulate the search problem in Eq. 1 as a gradient-based differentiable optimization. We term this approach *Differentiable Textual Optimization (DTO)*, which differentiates reward over token space for progressive response improvement.

**Objective.** Our overall goal is to refine a given sequence of tokens $\boldsymbol{y}^{(0)}$ so as to maximize the reward. However, directly maximizing $r(\boldsymbol{y}|\boldsymbol{x})$ risks *reward hacking* (Pan et al., 2022), as the optimization trajectory of $\boldsymbol{y}$ may drift away from the distribution under which $r(\boldsymbol{y}|\boldsymbol{x})$ is well-calibrated – typically the prior distribution induced by $\pi_{LLM}$. To mitigate this, we constrain $\boldsymbol{y}$ to remain within the high-density region of $\pi_{LLM}$. Concretely, we regularize the log-likelihood of $\boldsymbol{y}$, thereby penalizing

deviations from the distribution represented by the language model. The resulting objective function to be minimized is given by:

$$\mathcal{L}(\boldsymbol{y}) := -\lambda r(\boldsymbol{y}|\boldsymbol{x}) - \log \pi_{LLM}(\boldsymbol{y}|\boldsymbol{x}), \tag{2}$$

where $\lambda > 0$ is a hyper-parameter to balance the reward value and the regularization term. Intuitively, Eq. 2 seeks a response $\boldsymbol{y}$ that not only achieves a high reward but also maintains fluency and faithfulness in natural language (Kumar et al., 2021; Qin et al., 2022; Yuan et al., 2025). To estimate the log-likelihood $\log \pi_{LLM}(\boldsymbol{x}|\boldsymbol{y})$, we decompose it sequentially from left to right, which results in the next-token prediction loss: $\log \pi_{LLM}(\boldsymbol{y}|\boldsymbol{x}) = \sum_{i=1}^{|\boldsymbol{y}|} \boldsymbol{y}_i^\top \log \text{Cat}\left(\pi_{LLM}(\cdot|\boldsymbol{y}_{\leq i-1}, \boldsymbol{x})\right)$.

**Parameterization.** The token space of $\boldsymbol{y}$ is a discrete where gradients cannot directly operate. Therefore, we propose to parameterize the tokens via the underlying logit vectors used to sample them. At the initialization stage, we use the LLM-generated logits to initialize $\boldsymbol{z}^{(0)} \in \mathbb{R}^{|\boldsymbol{y}^{(0)}| \times |\mathcal{V}|}$. During optimization, we use Gumbel-softmax straight-through trick to parameterize $\boldsymbol{y}_i^{(t)} = \boldsymbol{\delta}_{\arg\max_{j \in |\mathcal{V}|} \boldsymbol{z}_{ij}^{(t)}} + \text{softmax}(\boldsymbol{z}_i^{(t)}/\tau) - \text{StopGrad}(\text{softmax}(\boldsymbol{z}_i^{(t)}/\tau))$ (Bengio et al., 2013; Jang et al., 2016), where $\boldsymbol{\delta}_i$ denotes the $i$-th canonical basis and $\tau > 0$ is the temperature coefficient. By this means, gradient descent can be equivalently performed on the space of $\boldsymbol{z}^{(t)}$ as: $\boldsymbol{z}^{(t+1)} = \boldsymbol{z}^{(t)} - \eta \nabla_{\boldsymbol{z}} \mathcal{L}(\boldsymbol{z}^{(t)})$.

As we will demonstrate in Sec. 4, the gradient of $\mathcal{L}$ propagates information bidirectionally. Preceding tokens act as a regularizer on successive tokens, enforcing consistency with the autoregressive generation process, while trailing tokens propagate outcome-level reward signals and full-sequence alignment back to earlier tokens through attention. This implements a closed-loop control of earlier predictions influencing subsequent decoding steps, thereby capturing the key recursive structure of reasoning characterized in Eq. 1. Furthermore, in Sec. 4, we establish a close connection between DTO, which optimizes directly in the sample space, and policy-optimization (e.g., PPO) that operate in the parameter space. We show that DTO provably shifts the drawn samples toward the reward-maximizing distribution induced by the original policy.

## 3.2 ITERATIVE DECODING WITH DTO

In this section, we elaborate on the detailed iterative generation process with DTO integrated for policy improvement. Akin to autoregressive decoding, $\nabla$-Reasoner generates the full response token by token. The sampling of each token consists of the following two steps:

**Policy Improvement via DTO.** Starting from a prefix $\boldsymbol{x} \in \mathcal{V}^*$, we let the LLM $\pi_{LLM}$ generate a continuation sequence $\boldsymbol{y}^{(0)}$ along with its pre-softmax logits $\boldsymbol{z}^{(0)}$. We then apply the DTO algorithm to optimize $\boldsymbol{z}^{(0)}$ for $T$ steps, yielding refined logits $\widetilde{\boldsymbol{z}}$. The logits corresponding to the first token are treated as the improved policy for predicting the immediate next token, intentionally adjusted to yield higher reward when used to generate the continuing responses. Accordingly, we resample the next token from this updated policy: $\widetilde{\boldsymbol{y}}_1 \sim \text{softmax}(\widetilde{\boldsymbol{z}}_1/\tau)$.

**Rejection Sampling.** Once a new next-token candidate $\widetilde{\boldsymbol{y}}_1$ is obtained, we first compare it with the initial prediction $\boldsymbol{y}_1$. If $\widetilde{\boldsymbol{y}}_1 = \boldsymbol{y}_1$, no effective policy update occurs, and we proceed directly to the next token for policy refinement. If $\widetilde{\boldsymbol{y}}_1 \neq \boldsymbol{y}_1$, we perform an additional rollout conditioned on $\widetilde{\boldsymbol{y}}_1$ as the next token, yielding a new response $\widetilde{\boldsymbol{y}}$. Both $\boldsymbol{y}$ and $\widetilde{\boldsymbol{y}}$ are then evaluated under the reward function, and the token that yields a full response with the higher reward is retained.

**Test-Time Scaling.** We scale computation in $\nabla$-Reasoner along two axes: (1) increasing the number of gradient update steps used by DTO to refine the policy, and (2) performing rejection sampling among rollouts yielded by the original and updated policy. Comparatively, allocating additional compute to gradient-based optimization is not only more effective in incorporating reward signals into the policy, but also more efficient than purely autoregressive decoding. This efficiency arises because computing the full-sequence gradient $\nabla_{\boldsymbol{z}} \mathcal{L}$ leverages the parallel execution of transformers: a single gradient step propagates updates across all tokens within one model call, whereas a standard autoregression generates only a single token per model call. As we will show in Sec. 5.4, sampling from the policy refined by DTO yields a significantly higher chance of reward improvement.

### 3.3 Accelerating ∇-Reasoner

The naive implementation of $\nabla$-Reasoner is inefficient due to two primary bottlenecks: (1) decoding each token requires a full optimization procedure, where each step involves backpropagation through two large models; and (2) generating a single token requires an additional full rollout. In this section, we demonstrate that $\nabla$-Reasoner is amenable to several strategies that significantly accelerate both optimization and generation, while adaptively allocating compute to the tokens that matter most.

**Gradient Caching.** The gradient backpropagated to the logits $\boldsymbol{z}$ can be decomposed via the chain rule as: $\nabla_{\boldsymbol{z}} \mathcal{L} = \frac{\partial \boldsymbol{z}}{\partial \boldsymbol{y}} \frac{\partial \mathcal{L}}{\partial \boldsymbol{y}}$, wherein the term $\frac{\partial \mathcal{L}}{\partial \boldsymbol{y}}$ dominates the computational cost, since it requires a full forward and backward pass through both the language model and the reward model. However, we observe that $\boldsymbol{y}$ – the one-hot vectors indicating the maximal entries in the soft logits $\boldsymbol{z}$ – changes infrequently as optimization proceeds. Exploiting this property, we cache the gradient $\frac{\partial \mathcal{L}}{\partial \boldsymbol{y}}$ once computed, and reuse it until the maximal entries of $\boldsymbol{z}$ flip. In our implementation, we retain the cached gradient $\boldsymbol{g}_i = \frac{\partial \mathcal{L}}{\partial \boldsymbol{y}_i}$ for every $i \in [|\boldsymbol{y}|]$ whenever $\boldsymbol{y}$ is updated, and otherwise recover it efficiently using the surrogate loss $\mathcal{L}_{cache} = \sum_{i=1}^{|\boldsymbol{y}|} \boldsymbol{y}_i^\top \boldsymbol{g}_i$ to recover the saved gradients $\{\boldsymbol{g}_i\}_{i \in [|\boldsymbol{y}|]}$ whenever $\boldsymbol{y}$ remains unchanged from the previous iteration. See Algorithm 4 in Appendix B.

**Rollout Trajectory Reusing.** We further note that the rollout strategy can be improved to reduce unnecessary computation and better leverage the KV cache. In the naive implementation (Sec. 3.2 or Algorithm 1), $\nabla$-Reasoner generates a sequential rollout to optimize the next-token prediction policy for every decoding step. However, the rollout trajectory, including both tokens and logits, continuing from the previously accepted token can be directly reused as the rollout for the subsequent token. In Algorithm 1, we skip the rollout at the beginning of each step and reuse $\boldsymbol{y}_{>2}$ and $\boldsymbol{z}_{\geq 2}$ as the rollout for the next token if the resampled token $\widetilde{\boldsymbol{y}}$ is rejected; otherwise, we continue with $\widetilde{\boldsymbol{y}}$ and $\widetilde{\boldsymbol{z}}$ for the next step. We also limit the total number of rollouts as $N_{max}$. Once the total number of rollouts exceeds the $N_{max}$, we terminate the iterative policy refinement and generate the remaining tokens using standard autoregressive decoding. See more details in Algorithm 3 in Appendix B.

**Confidence- and Gradient-Guided Token Selection.** Running DTO to optimize the policy at every decoding step can result in redundant computation. We observe that token logits with either high confidence (see Appendix C.3) or small gradients are unlikely to be modified under DTO. To address this, we introduce two selection criteria, *entropy-based* and *gradient-based* to determine which tokens should undergo policy refinement. Specifically, we define two hyperparameters, $\epsilon_{ent}$ and $\epsilon_{grad}$. DTO is applied only when the entropy of the token logits satisfies $H(\boldsymbol{z}_1) > \epsilon_{ent}$ and the gradient magnitude exceeds $\|\nabla_{\boldsymbol{z}_1} \mathcal{L}\|_2 > \epsilon_{grad}$, where $H(\cdot)$ denotes the entropy of a categorical distribution. We refer readers to Algorithm 4 in Appendix B for more details.

## 4 Theoretical Analysis

**Interpretation of Gradient Updates.** We analyze the gradient of $\mathcal{L}$ to reveal how DTO updates the response. The derivatives in terms of the $l$-th token $\partial \mathcal{L} / \partial \boldsymbol{x}_l$ under loss Eq. 2 can be decomposed as follows:

$$\frac{\partial \mathcal{L}}{\partial \boldsymbol{y}_i} = -\underbrace{\log \mathrm{Cat}\left(\pi_{LLM}(\cdot | \boldsymbol{y}_{\leq i-1}, \boldsymbol{x})\right)}_{\boldsymbol{\delta}_{prefix}} - \underbrace{\sum_{j=i+1}^{|\boldsymbol{y}|} \frac{\partial \log \mathrm{Cat}\left(\pi_{LLM}(\cdot | \boldsymbol{y}_{\leq j-1}, \boldsymbol{x})\right)}{\partial \boldsymbol{y}_i} \boldsymbol{x}}_{\boldsymbol{\delta}_{postfix}} - \lambda \underbrace{\frac{\partial r(\boldsymbol{y}|\boldsymbol{x})}{\partial \boldsymbol{y}_i}}_{\boldsymbol{\delta}_{reward}}.$$

We defer the derivation to Appendix C.1. The first term, $\boldsymbol{\delta}_{\text{prefix}}$, updates the token $\boldsymbol{x}_i$ based on its preceding context, aligning the next-token policy with the autoregressive prediction probabilities produced by the language model. The second term, $\boldsymbol{\delta}_{\text{postfix}}$, propagates information from subsequent tokens through the attention mechanism, encouraging global consistency with respect to its future context. Finally, the reward gradient $\boldsymbol{\delta}_{\text{reward}}$ provides a sequence-level signal, transmitting information from later tokens to $\boldsymbol{y}_i$ via attention. As highlighted by Bachmann & Nagarajan (2024), the order of generation plays a crucial role in complex reasoning or algorithmic tasks. Pure left-to-right generation can fall short of *error accumulation*, making it insufficient for intricate logical reasoning processes. An ideal decoding method for reasoning should allow for iterative refinement of the reasoning chain in both forward and backward directions (Yao et al., 2024; Hao et al., 2023).

**Inference-Time Gradient Descent is "Deamortized" PPO.** We theoretically establish the connection between the test-time textual optimization and parametric RL-based training. RLHF (Schulman et al., 2017; Ouyang et al., 2022) and RLVR (Guo et al., 2025; Shao et al., 2024) have been demonstrated to be particularly effective for mathematical reasoning tasks (Wang et al., 2023a; Zhao et al., 2023; Dong et al., 2024; Shao et al., 2024). The primary objective of RL is to fine-tune a pre-trained LLM using RL algorithms, ensuring that its responses to specific prompts maximize a given reward function. Among various RL algorithms, KL-regularized approaches, such as Proximal Policy Optimization (PPO) (Schulman et al., 2017), have been widely adopted in practice. In this section, we uncover the hidden connection between PPO and our proposed DTO. Formally, let $\rho : \mathcal{V}^* \to \mathbb{R}$ represent an LLM to be aligned with the reward function $r$, initialized from the pre-trained policy $\pi_{LLM}$. PPO optimizes for $\rho$ by minimizing the following functional objective defined over the space of distributions:

$$\mathcal{L}_{PPO}(\rho) := - \mathbb{E}_{\boldsymbol{y} \sim \rho}[\lambda r(\boldsymbol{y})] + D_{\mathrm{KL}}(\rho \| \pi_{LLM}), \tag{3}$$

where the first expectation term estimates the expected reward, while in the second term, the KL-divergence regularizes the distributional discrepancy between $\rho$ and $\pi_{LLM}$. Assuming LLMs' input domain can be extended to the ambient space beyond discrete vocabularies, then we can show the relation between (stochastic) gradient flow of Eq. 2 and functional solution to PPO:

**Theorem 4.1.** *Suppose $\{\rho^t\}_{t \geq 0}$ denotes the Wasserstein gradient flow minimizing Eq. 3 in the distribution space with boundary conditions $\rho^0 = \pi_{LLM}$ and $\rho^\infty = \rho^* = \arg\min_\rho \mathcal{L}_{PPO}(\rho)$. Then we can draw samples from $\rho^*$ by first initializing $\boldsymbol{x}^0 \sim \pi_{LLM}$ and simulating a trajectory $\{\boldsymbol{x}^t\}_{t \geq 0}$ following the stochastic gradient flow of Eq. 2: $\frac{d\boldsymbol{x}^t}{dt} = -\nabla \mathcal{L}(\boldsymbol{x}^t) + \sqrt{2}\boldsymbol{\epsilon}_t$, where $\{\boldsymbol{\epsilon}_t \in \mathcal{N}(\boldsymbol{0}, \boldsymbol{I})\}_{t \geq 0}$ are Brownian motions.*

Theorem 4.1 is proved in Appendix C.2. Theorem 4.1 shows that instead of optimizing the entire policy to satisfy the reward function w.r.t. Eq. 3, there exists a trajectory driven by the gradients of Eq. 2 on the sample space that can directly generate samples from the optimal distribution minimizing Eq. 3. Pre-training scaling and test-time scaling can be unified and interpreted through Theorem 4.1 as two complementary forms of statistical inference: parametric and non-parametric (particle-based) inference (Liu & Wang, 2016; Chen et al., 2018). The pre-training stage corresponds to parametric inference: a global parameter is optimized to minimize the overall loss across a dataset, amortizing the cost of individual samples into a shared parameter. Increasing the size of this parameter space enhances the model's representational capacity, thereby reducing the average cost per sample. In contrast, test-time scaling via DTO is analogous to non-parametric inference, which performs optimization in the sample space, treating each sample as an independent "particle" that minimizes its own cost. This allows for fine-grained adaptation at the individual sample level. The Wasserstein gradient flow provides a mathematical framework to describe the relationship between the dynamics of measures (global distributions) and individual samples, thereby bridging the conceptual gap between pre-training scaling and test-time scaling.

## 5 EXPERIMENTS

### 5.1 RESULTS ON MATH REASONING

**Experiment Details.** Tab. 1 compares our test-time method, $\nabla$-Reasoner, against a variety of baselines. We benchmark its performance against other test-time approaches, including greedy decoding, Self-Consistency (SC) (Xie et al., 2024), Best-of-N (BoN) (Stiennon et al., 2020), tree-search based methods: Tree-of-Thought (ToT) (Yao et al., 2024) and Reasoning via Planning (RAP) (Hao et al., 2023), and the iterative refinement approach: TPO (Li et al., 2025). Additionally, we include training-based methods such as Supervised Fine-Tuning (SFT) and GRPO (Guo et al., 2025) for a comprehensive comparison. We evaluate two model families, Qwen-2.5-math (Yang et al., 2024a) and Llama-3.1 (Grattafiori et al., 2024), on four representative mathematical reasoning benchmarks: MATH-500 (Hendrycks et al., 2021), AIME24, AIME25, and AMC (Li et al., 2024). We leverage reward models from the Skywork-V2 family (Liu et al., 2025): for Skywork-V2-Qwen-4B for Qwen-based models and Skywork-V2-Llama-8B for Llama family models. For BoN and SC, we let $N = 8$ to match $N_{max} = 8$ used in our methods. For TPO, we set the number of samples per step as $N_{samples} = 2$ and the number of refinement steps as $N_{refine} = 2$. For ToT and RAP, we adopt the default hyperparameters in Hao et al. (2024) to yield meaningful results. Please refer to Appendix D for more experimental details.

Table 1: Accuracy (%) on math reasoning datasets compared with baseline methods, including both test-time and training-time approaches. We skip results on AIME datasets for Llama-3.1-8B as it is incapable of generating reasonable performance. We mark the best performer in **bold** and the runner-up with underline. Our method outperforms all test-time baselines and even achieves performance on par with the training-based methods (SFT and GRPO), respectively.

| Models | Methods | MATH-500 | AMC | AIME24 | AIME25 |
|---|---|---|---|---|---|
| Qwen-2.5-7B | Greedy decoding | 43.8 | 33.0 | 6.7 | 6.7 |
| | SC (Xie et al., 2024) ($N = 8$) | 69.8 | 49.4 | 22.5 | 20.0 |
| | BoN (Stiennon et al., 2020) ($N = 8$) | 70.2 | 50.1 | 22.5 | 13.3 |
| | ToT (Yao et al., 2024) | 57.8 | 42.4 | 6.7 | 10.0 |
| | RAP (Hao et al., 2023) | 68.6 | 50.1 | 18.3 | 14.2 |
| | SFT (Ouyang et al., 2022) | 65.8 | 36.4 | 6.3 | 11.7 |
| | GRPO (Guo et al., 2025) | 70.8 | **52.8** | 20.8 | **16.7** |
| | $\nabla$-Reasoner ($N_{max} = 8$) | **71.0** | 51.5 | **23.3** | 15.0 |
| Qwen-2.5-7B-Instruct | Greedy decoding | 71.2 | 43.0 | 5.3 | 7.5 |
| | SC (Xie et al., 2024) ($N = 8$) | 76.6 | 55.5 | 25.0 | **22.5** |
| | BoN (Stiennon et al., 2020) ($N = 8$) | 77.8 | 55.9 | 22.5 | 18.3 |
| | ToT (Yao et al., 2024) | 75.4 | 48.2 | 20.0 | 18.3 |
| | RAP (Hao et al., 2023) | 80.2 | 54.6 | 1.6 | 12.5 |
| | TPO (Li et al., 2025) | 77.6 | 55.9 | 6.7 | 11.1 |
| | $\nabla$-Reasoner ($N_{max} = 8$) | 80.4 | 56.8 | 26.6 | 20.0 |
| Llama-3.1-8B-Instruct | Greedy decoding | 40.6 | 19.3 | - | - |
| | SC (Xie et al., 2024) ($N = 8$) | 54.8 | 25.7 | - | - |
| | BoN (Stiennon et al., 2020) ($N = 8$) | 52.2 | 26.1 | - | - |
| | ToT (Yao et al., 2024) | 50.2 | 25.6 | - | - |
| | RAP (Hao et al., 2023) | 55.4 | 25.8 | - | - |
| | SFT (Ouyang et al., 2022) | 46.6 | 20.2 | - | - |
| | $\nabla$-Reasoner ($N_{max} = 8$) | **55.8** | **28.9** | - | - |

**Performance Comparison.** Our method shows superior performance across all models and benchmarks on test-time methods. We even reach comparable performance with training-based methods. Specifically, with the Qwen-2.5-7B base model, our approach achieves the highest scores among test-time methods on MATH-500 (71.0%) and AIME24 (23.3%), and remains highly competitive with the training-based GRPO method (trained with 35k examples). The advantage is even more pronounced with the instruction-tuned Qwen-2.5-7B-Instruct and Llama-3.1-8B-Instruct, where our method again leads all other approaches on all benchmarks. Notably, on Qwen-2.5-7B-Instruct, our method scores 80.4% on MATH-500 and 56.8% on AMC, and on Llama-3.1-8B-Instruct, our method achieves 55.8% on MATH-500 and 28.9% on AMC. In the meantime, we also provide an example to show how $\nabla$-Reasoner works in real-world scenarios in Appendix D.4.

## 5.2 COST COMPARISON

We note that $\nabla$-Reasoner can exhibit an efficiency advantage by utilizing compute more effectively. This stems from $\nabla$-Reasoner's ability to leverage parallelized attention execution to compute gradients and update the decoding-time policy on the fly, which often leads to a small practical runtime. To

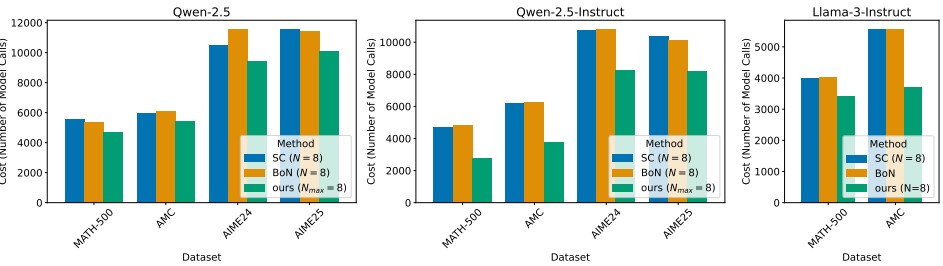

Figure 3: A comparison of computational cost, measured by the number of model calls. Our method reduces costs by up to 40.2% compared to baselines.

capture this advantage, we propose using the number of model calls as a surrogate theoretical efficiency metric. This metric counts both a single recurrent computation and a parallel forward pass as one model call. The choice is motivated by the practical observation that, under ideal attention parallelization, a single model call – regardless of whether it is executed recurrently or in parallel – can incur comparable wall-clock cost. Thus, it serves as a unifying metric that reflects algorithmic complexity under idealized system-level and hardware optimizations, while mitigating discrepancies arising from implementation details (e.g., compatibility with serving engines).

Fig. 3 compares the computational cost of our method with several baselines. For fair comparison, we set the number of generated samples to 8 for all approaches ($N = 8$ for SC and BoN, and $N_{max} = 8$ for our $\nabla$-Reasoner). Our method delivers superior performance at a significantly lower cost than SC and BoN. For instruction-tuned models, it reduces the number of model calls by up to 40.2%, while for base models, it outperforms all baselines using only about 90% of this metric. The reason for the reduced cost is twofold: (1) with confidence- and gradient-guided selection, rollouts usually start from the middle of the sequence, instead of from the beginning as BoN and SC do; (2) the optimization cost with gradient caching remains lightweight while our DTO enables efficient parallelizable execution of transformers and revision of tokens. These results imply that $\nabla$-Reasoner has stronger prospects for achieving inference efficiency. Unlike trial-and-error–based test-time scaling methods (e.g., BoN), which repeatedly resample outputs without guidance, our method updates the decoding policy in a targeted and strategic fashion.

## 5.3 TEST-TIME SCALING LAW

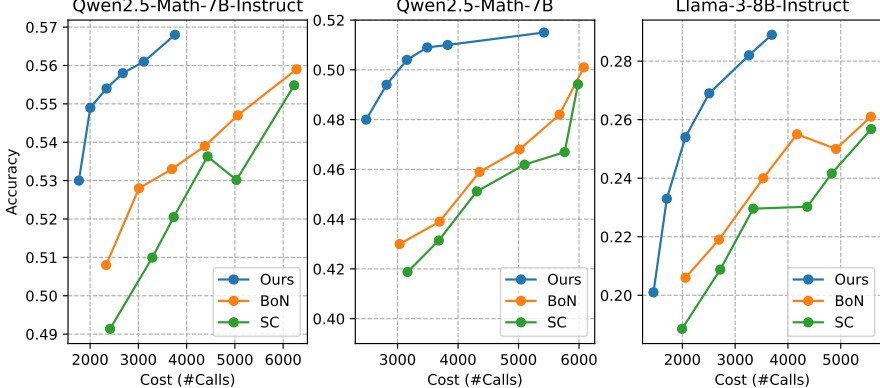

Figure 4: Test-time scaling curves comparing our method with BoN and SC. We change the number of samples $N$ for BoN and SC and number of rollouts $N_{max}$ for our method. The results show $\nabla$-Reasoner achieves superior performance with reduced cost across multiple models.

We present our test-time scaling curves in Fig. 4, comparing our method against Best-of-N (BoN) and Self-Consistency (SC). The figure plots accuracy against computational cost, showing how each method's performance scales as more resources are used. As is evident across all models, our method's curve consistently lies above the baselines. This indicates that for any given computational budget (number of calls), our approach achieves a higher accuracy. These results demonstrate that $\nabla$-Reasoner offers a superior trade-off between performance and computational cost, establishing a more efficient frontier compared to these sample-heavy techniques.

## 5.4 ALGORITHM ANALYSIS

**Dependencies on Reward Models.** Our approach relies on the gradient signal from the reward model to optimize policy at test time. To study the dependency on the quality of reward models, we further evaluate our approach on Qwen2.5-Math-7B-Instruct paired with the larger Skywork-Reward-V2-Qwen3-8B reward model. We note that our original choice Skywork-Reward-V2-Qwen3-4B is a smaller and weaker reward model according to

Table 2: Ablation study on reward model choice.

| Model | Skywork-Qwen3-4B | Skywork-Qwen3-8B |
|---|---|---|
| MATH-500 | 80.4 | 80.8 (+0.4) |
| AMC | 56.8 | 57.1 (+0.3) |

the RewardBench (Malik et al., 2025). According to the Tab. 2, the performance gap between the 4B and 8B variants remains consistently below 1 point across both MATH-500 and AMC. This indicates that using a smaller reward model does not lead to significant performance degradation compared with the larger, stronger version. This justifies our original choice (in Tab. 1) and further suggests smaller reward models may be preferable for improving efficiency.

Table 3: Analysis of rejection rate (%) in rejection sampling. We set $N = 8$ for the BoN baseline and also set $N_{max} = 8$ for our $\nabla$-Reasoner. The theoretical rejection rate of the baseline is 66.0%.

| Model | Baseline | $\nabla$-Reasoner |
|---|---|---|
| Qwen-2.5 | 65.9 | **32.8** |
| Qwen-2.5-Instruct | 66.5 | **28.9** |
| Llama-3-Instruct | 66.9 | **40.1** |

**Rejection Rate Analysis.** As described in Sec. 3.2 and Algorithm 1, $\nabla$-Reasoner first applies DTO to directly optimize the policy in the logit space, and then compares a continuation $\widetilde{y}$ generated from a resampled token with the original rollout sequence $y$. The token sampled from the optimized policy is adopted only if it yields a continuation with a higher reward. Henceforth, it becomes essential to quantify the acceptance rate of tokens drawn from the optimized policy in order to justify the effectiveness of DTO.

To this end, we measure the *rejection rate*, defined as the percentage of candidates produced by DTO that are rejected for failing to improve the reward. For comparison, we also evaluate this metric on a baseline that performs rejection sampling without DTO, which is equivalent to BoN. Theoretically, performing rejection sampling $N$ times over an identical distribution yields a rejection rate of $1 - (\sum_{k=1}^{N} 1/k)/N$ that converges to one as $N \to \infty$. For $N = 8$, the expected rejection rate is approximately 66.0%. We report our measured rejection rate in Tab. 3. We report the empirical rejection rates in Tab. 3. The results show that rejection sampling without DTO closely matches the theoretical prediction, while rejection sampling with DTO significantly reduces the rejection rate (by up to 30%). This confirms that DTO is effective in improving the next-token policy, producing tokens that lead to continuations with higher rewards.

# 6 CONCLUSION AND LIMITATIONS

We presented $\nabla$-Reasoner, an inference-time reasoning framework that introduces Differentiable Textual Optimization (DTO) to refine token logits via gradient-based optimization. By combining gradient signals from both the LLM likelihood and a reward model, $\nabla$-Reasoner enables more effective policy improvement than zeroth-order search methods, while incorporating rejection sampling and speedup techniques to boost effectiveness and efficiency. Theoretically, we show that aligning with a reward function is equivalent to gradient-based optimization in the sample space. $\nabla$-Reasoner delivers substantial performance gains over base models while consistently reducing computation cost, illustrating a sharper and more efficient scaling paradigm for LLM reasoning.

**Limitations.** In line with previous observations (Yue et al.), the performance of $\nabla$-Reasoner appears to remain bounded by the capabilities of the underlying base model and reward model, particularly under limited computation. The base and reward models are required to share the same vocabulary to allow for end-to-end logit optimization. Moreover, integrating $\nabla$-Reasoner into efficient LLM serving pipelines requires more careful system co-design to incorporate test-time gradient descent.

ACKNOWLEDGMENTS

Authors thank Shibo Hao, Junbo Li, and Sina Alemohammad for helpful discussions. This work was supported in part by NSF Awards 2145346 (CAREER) and the NSF AI Institute for Foundations of Machine Learning (IFML). It was also supported by computing support on the Vista GPU Cluster through the Center for Generative AI (CGAI) and the Texas Advanced Computing Center (TACC) at The University of Texas at Austin. Peihao Wang is in part supported by Google PhD Fellowship in Machine Learning and ML Foundations. Ruisi Cai is in part supported by NVIDIA Graduate Fellowship. Zhen Wang is supported by OpenAI Research Grant and Gordon and Betty Moore Foundation Postdoctoral Fellowship.

ETHICS STATEMENT

This research primarily concentrates on developing inference algorithms to enhance reasoning in large language models (LLMs), with a particular emphasis on mathematical reasoning. It relies solely on extant LLMs and does not involve training, fine-tuning model weights, or creating new LLMs. As a result, the work does not raise any novel domain-specific ethical considerations or societal impacts beyond those already well-documented in relation to large-scale language models more broadly.

REPRODUCIBILITY STATEMENT

We include pseudocode plus a detailed version in both the main text and Appendix B. Complete derivations and proofs are provided in Appendix C. Additionally, Appendix D contains the full list of hyperparameters, datasets, and model checkpoints required to reproduce our experimental results.

THE USE OF LARGE LANGUAGE MODELS

Large language models are used solely for sentence-level proofreading. All research ideation and paper writing were originally carried out by the authors.

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

## A  OTHER RELATED WORK

**Scaling LLM Reasoning at Inference Time.**   The paradigm of scaling inference-time compute has emerged as a powerful strategy to amplify the reasoning capabilities of LLMs. Chain-of-thought (CoT) prompting Wei et al. (2022b) pioneered this direction by eliciting multi-step reasoning explicitly. Subsequent works scale inference-time compute by sampling more reasoning chains or exploring larger reasoning spaces. For instance, Best-of-N (BoN) (Stiennon et al., 2020; Nakano et al., 2021) and Self-consistency (SC-CoT) Wang et al. (2022) repeatedly sample multiple chains and select the best one using heuristics such as predefined rewards or the consistency. More advanced approaches have introduced various search algorithms to efficiently explore the reasoning space Yao et al. (2024); Besta et al. (2024); Hao et al. (2023; 2024). Tree-of-thought (ToT) Yao et al. (2024) formulates reasoning as a tree search problem, where the LLM generates thoughts while heuristic rewards guide tree traversal. RAP Hao et al. (2023; 2024) employs Monte Carlo Tree Search (MCTS) to strategically balance exploration and exploitation in the search space. Recent work by Snell et al. (2024) further formalizes the empirical benefits of test-time compute scaling. In contrast to these sampling and search-based methods, our work proposes gradient-based optimization to traverse the reward landscape more efficiently, bypassing the trial-and-error inefficiencies of sampling approaches. Notably, our method is orthogonal to recent RL-trained methods for test-time scaling, such as OpenAI's o1 and DeepSeek's R1 – our method can be directly applied to these models to refine their long CoT inference, though we leave this exploration to future work.

**Inference-time Constrained Decoding.**   Constrained decoding is a traditional problem to study in text generation, with popular applications including controllable text generation and preference alignment. Recently, inference-time alignment has been heavily studied to align LLMs with human values as alignment rewards Li et al. (2023); Huang et al. (2024); Khanov et al. (2024). Most of them formulate the problem as a reward-guided search process. More relevantly, controllable text generation has leveraged energy-based models (EBMs) (Kumar et al., 2021; Qin et al., 2022; Kumar et al., 2022; Mireshghallah et al., 2022; Liu et al., 2023b; Yuan et al., 2025) to relax discrete text sequences into continuous spaces, which enables gradient-based optimization (e.g., Langevin dynamics) to steer generation toward objectives defined by arbitrary energy functions. While sharing similar reward-constrained decoding principles and gradient-based methodologies, end-to-end sequence optimization is notoriously unstable, often producing broken sequences or failing to converge, which limits its applicability to reasoning tasks. In contrast, our approach introduces an efficient and novel gradient-based framework integrated with the iterative decoding, tailored for reasoning tasks.

**Prompt Optimization.**   A distinct but related line of research focuses on optimizing prompts rather than decoding sequences. Established principles from the above sections apply here as well. Search-based methods Zhou et al. (2022); Wang et al. (2023b); Pryzant et al. (2023); Yuksekgonul et al. (2024) iteratively refine the prompts through automated trial-and-error, while gradient-based approaches include directly optimizing soft prompts Li & Liang (2021); Lester et al. (2021), or searching for discrete prompts via gradients Shin et al. (2020); Shi et al. (2022); Wen et al. (2024). Although these techniques share conceptual similarities with our method, esp. the gradient-based optimization methods, they operate on the input (prompt) space rather than the output (reasoning chain) space. Our work focuses on optimizing reasoning trajectories directly, complementing rather than competing with prompt optimization methods – future work could explore synergies between the two paradigms.

**Continuous Latent Space Reasoning.** Another line of research explores reasoning in a continuous latent space, bypassing discrete token-level operations. These approaches typically perform iterative refinement on the model's hidden states to solve reasoning problems. For example, some methods frame reasoning as an energy minimization process (Du et al., 2022) or a fixed-point iteration problem within the latent space, which can be parallelized for efficiency (Wu et al., 2025). Others propose increasing test-time compute by applying recurrent updates to latent representations, effectively deepening the model's computation on-the-fly (Geiping et al., 2025). This paradigm has also been supported by specialized pre-training objectives that encourage models to "ponder" in a continuous space (Zeng et al., 2025) and has received theoretical analysis under the lens of continuous chain-of-thought (Zhu et al., 2025). While conceptually related in their use of iterative refinement, these methods are fundamentally different from ours. They operate within the LLM's latent space, modifying internal hidden representations. In contrast, our work, $\nabla$-Reasoner, performs optimization

---

**Algorithm 3** $\nabla$-Reasoner: Decoding with DTO

---

**Require:** Prompt $x$, language model $\pi_{LLM}$, reward model $r$, stop criteria StopCriteria, and the maximal number of rollouts $N_{max}$.
1: $\boldsymbol{y}, \boldsymbol{z} \sim \pi_{LLM}(\cdot|\boldsymbol{x})$
2: $N_r \leftarrow 1$
3: **repeat**
4:  **if** $H(\boldsymbol{z}_1) \geq \epsilon_{ent}$ and $\|\nabla_{\boldsymbol{z}_1}\mathcal{L}\|_2 \geq \epsilon_{grad}$ **then**         $\triangleright$ Token selection (Sec. 3.3).
5:   $\widetilde{\boldsymbol{z}} \leftarrow \mathrm{DTO}(\boldsymbol{x}, \boldsymbol{z}, \pi_{LLM}, r)$.      $\triangleright$ Policy refinement with DTO (Sec. 3.1).
6:  **end if**
7:  $\widetilde{\boldsymbol{y}}_1 \sim \mathrm{softmax}(\widetilde{\boldsymbol{z}}_1/\tau)$.
8:  **if** $\widetilde{\boldsymbol{y}}_1 \neq \boldsymbol{y}_1$ **then**
9:   $\widetilde{\boldsymbol{y}}, \widetilde{\boldsymbol{z}} \sim \pi_{LLM}(\cdot|\widetilde{\boldsymbol{y}}_1, \boldsymbol{x})$
10:   $N_r \leftarrow N_r + 1$
11:   **if** $r(\widetilde{\boldsymbol{y}}, \widetilde{\boldsymbol{y}}_1|\boldsymbol{x}) > r(\boldsymbol{y}|\boldsymbol{x})$ **then**       $\triangleright$ Rejection sampling (Sec. 3.2)
12:    $\boldsymbol{x} \leftarrow \mathrm{concat}[\boldsymbol{x}, \widetilde{\boldsymbol{y}}_1]$
13:    $\boldsymbol{y} \leftarrow \widetilde{\boldsymbol{y}}, \boldsymbol{z} \leftarrow \widetilde{\boldsymbol{z}}$         $\triangleright$ Rollout reusing (Sec. 3.3)
14:    **continue**
15:   **end if**
16:  **end if**
17:  $\boldsymbol{x} \leftarrow \mathrm{concat}[\boldsymbol{x}, \boldsymbol{y}_1]$
18:  $\boldsymbol{y} \leftarrow \boldsymbol{y}_{\geq 2}, \boldsymbol{z} \leftarrow \boldsymbol{z}_{\geq 2}$         $\triangleright$ Rollout reusing (Sec. 3.3)
19:  **if** $N_r \geq N_{max}$ **then**             $\triangleright$ Early stop (Sec. 3.3).
20:   $\boldsymbol{x} \leftarrow \mathrm{concat}[\boldsymbol{x}, \boldsymbol{y}]$
21:   **break**
22:  **end if**
23: **until** StopCriteria($\boldsymbol{x}$)
24: **return** $\boldsymbol{x}$

---

directly in the output space. We manipulate the token logits, a continuous relaxation of the discrete vocabulary, guided by reward gradients. This direct textual optimization allows us to refine the reasoning chain itself at inference time without altering the base model's internal forward pass or requiring any specialized training, uniquely positioning our method as a post-hoc, gradient-based search algorithm over the sequence space.

# B IMPLEMENTATION

## B.1 PSEUDOCODE

In Algorithms 1 and 2, we present a basic implementation for $\nabla$-Reasoner. Below we give a detailed pseudocode for a full version with all acceleration techniques integrated (Sec. 3.3). In Alg. 3, we list the complete version of iterative decoding with confidence- and gradient-guided token selection, rollout reusing, and early stop techniques. In Alg. 4, we present the full DTO algorithm with gradient caching. We note that these techniques significantly accelerate the decoding speed of $\nabla$-Reasoner, as demonstrated in Sec. 5.

## B.2 GENERALIZATION TO PROGRESS REWARD.

The reward function can take different forms: it may provide an *outcome reward* (Cobbe et al., 2021), offering an overall score for the entire response sequence, or a *process reward* (Lightman et al., 2023), which assesses individual intermediate steps and assigns a series of scores accordingly. Thus, beyond a single reward defined over the whole sequence, we denote the total reward as the sum of rewards obtained from different subsequences: $R(\boldsymbol{y}|\boldsymbol{x}) = \sum_{l=1}^{|\boldsymbol{y}|} r(\boldsymbol{y}_{\leq l}|\boldsymbol{x})$. In the case of an outcome reward, the reward is only assigned at the end of the response sequence, meaning $r(\boldsymbol{y}_{\leq l}|\boldsymbol{x}) = 0$ if $l < |\boldsymbol{y}|$. Conversely, when using a process reward, rewards are assigned incrementally, with $r(\boldsymbol{y}_{\leq l}|\boldsymbol{x}) \neq 0$ only if $\boldsymbol{y}_l$ is an end token of a thought (Xiong et al., 2024). Our framework can seamlessly incorporate a progress reward by replacing $r(\boldsymbol{y}|\boldsymbol{x})$ with this generalized version $R(\boldsymbol{y}|\boldsymbol{x})$.

---

**Algorithm 4** Differentiable Textual Optimization (DTO)

---

**Require:** Prefix $\boldsymbol{x}$, initial logits $\boldsymbol{z}$, language model $\pi_{LLM}$, reward model $r$, and the number of training steps $T$.

1: $\widehat{\boldsymbol{y}} \leftarrow$ None
2: $\boldsymbol{g}_1, \boldsymbol{g}_2, \cdots \leftarrow$ None
3: **while** $t < T$ **do**
4:     **for** every $i = 1, \cdots, |\boldsymbol{y}|$ **do**
5:         $j^* \leftarrow \arg\max_{j \in [|\mathcal{V}|]} \boldsymbol{z}_{ij}^{(t)}$
6:         $\boldsymbol{y}_i^{(t)} \leftarrow \boldsymbol{\delta}_{j^*} + \text{softmax}(\boldsymbol{z}_i^{(t)}/\tau) - \text{StopGrad}(\text{softmax}(\boldsymbol{z}_i^{(t)}/\tau))$
7:     **end for**
8:     **if** $\boldsymbol{y} \neq \widehat{\boldsymbol{y}}$ **then**
9:         $\mathcal{L}_{nll} = -\sum_{i=1}^{|\boldsymbol{y}^{(t)}|} \log \pi_{LLM}(\boldsymbol{y}^{(t)}|\boldsymbol{y}_{\leq i-1}^{(t)}, \boldsymbol{x})$
10:         $\mathcal{L}_{reward} = -r(\boldsymbol{y}^{(t)}|\boldsymbol{x})$.
11:         $\mathcal{L} = \mathcal{L}_{nll} + \lambda \mathcal{L}_{reward}$.             $\triangleright$ Eq. 2
12:         $\widehat{\boldsymbol{y}} \leftarrow \boldsymbol{y}$
13:         $\boldsymbol{g}_i \leftarrow \frac{\partial \mathcal{L}}{\partial \boldsymbol{y}_i^{(t)}}$ for every $i \in [|\boldsymbol{y}^{(t)}|]$       $\triangleright$ Gradient caching (Sec. 3.3)
14:     **else**
15:         $\mathcal{L} = \sum_{i=1}^{|\boldsymbol{y}^{(t)}|} \boldsymbol{g}_i^\top \boldsymbol{y}_i^{(t)}$         $\triangleright$ Surrogate loss with cached gradient (Sec. 3.3)
16:     **end if**
17:     $\boldsymbol{z}^{(t+1)} \leftarrow \boldsymbol{z}^{(t)} - \eta \nabla_{\boldsymbol{z}} \mathcal{L}$.
18:     $t \leftarrow t + 1$.
19: **end while**
20: **return** $\boldsymbol{z}^{(T)}$

---

## C   Deferred Theory

### C.1   Gradient Derivation of $\mathcal{L}$

We derive the gradients of Eq. 2 summarized as the following proposition. We consider the generalized reward function which is written as a summation over rewards defined over different subsequences (Sec. B.2).

**Proposition C.1.** *The gradient of loss function $\mathcal{L}(\boldsymbol{y}) = -\lambda \sum_{i=1}^{|\boldsymbol{y}|} r(\boldsymbol{y}_{\leq i}|\boldsymbol{x}) - \log \pi_{LLM}(\boldsymbol{y}|\boldsymbol{x})$ takes the form of $\frac{\partial \mathcal{L}(\boldsymbol{y})}{\partial \boldsymbol{y}_l} = \boldsymbol{\delta}_{prefix} + \boldsymbol{\delta}_{postfix} + \lambda \boldsymbol{\delta}_{reward}$ where:*

$$\boldsymbol{\delta}_{prefix} = -\log \text{Cat}\left(\pi_{LLM}(\cdot|\boldsymbol{y}_{\leq l-1}, \boldsymbol{x})\right), \tag{4}$$

$$\boldsymbol{\delta}_{postfix} = -\sum_{i=l+1}^{|\boldsymbol{y}|} \frac{\partial \log \text{Cat}\left(\pi_{LLM}(\cdot|\boldsymbol{y}_{\leq i-1}, \boldsymbol{x})\right)}{\partial \boldsymbol{y}_l} \boldsymbol{y}, \tag{5}$$

$$\boldsymbol{\delta}_{reward} = -\sum_{i=l}^{|\boldsymbol{y}|} \frac{\partial r(\boldsymbol{y}_{\leq i}|\boldsymbol{x})}{\partial \boldsymbol{y}_l}. \tag{6}$$

*Proof.* The proof is done by elementary derivative calculation. First of all, we write down the expanded expression of the loss function:

$$\mathcal{L}(\boldsymbol{y}) = -\lambda \sum_{i=1}^{|\boldsymbol{y}|} r(\boldsymbol{y}_{\leq i}|\boldsymbol{x}) - \sum_{i=1}^{|\boldsymbol{y}|} \log \pi_{LLM}(\boldsymbol{y}_i|\boldsymbol{y}_{\leq i-1}, \boldsymbol{x}) \tag{7}$$

$$= -\lambda \sum_{i=1}^{|\boldsymbol{y}|} r(\boldsymbol{y}_{\leq i}|\boldsymbol{x}) - \sum_{i=1}^{|\boldsymbol{y}|} \sum_{v \in [|\mathcal{V}|]} \boldsymbol{y}_{i,v} \log \pi_{LLM}(\boldsymbol{e}_v|\boldsymbol{y}_{\leq i-1}, \boldsymbol{x}) \tag{8}$$

For a specific token index $l \in [|\boldsymbol{y}|]$, we decompose the loss into five components:

$$
\mathcal{L}(\boldsymbol{y}) = \underbrace{-\lambda \sum_{i=1}^{l-1} r(\boldsymbol{y}_{\leq i}|\boldsymbol{x})}_{\Phi_1} \underbrace{-\lambda \sum_{i=l}^{|\boldsymbol{y}|} r(\boldsymbol{y}_{\leq i}|\boldsymbol{x})}_{\Phi_2} \underbrace{- \sum_{i=1}^{l-1} \sum_{v \in [|\mathcal{V}|]} \boldsymbol{y}_{i,v} \log \pi_{LLM}(\boldsymbol{e}_v|\boldsymbol{y}_{\leq i-1}, \boldsymbol{x})}_{\Pi_1} \tag{9}
$$

$$
\underbrace{- \sum_{v \in [|\mathcal{V}|]} \boldsymbol{y}_{l,v} \log \pi_{LLM}(\boldsymbol{e}_v|\boldsymbol{y}_{\leq l-1}, \boldsymbol{x})}_{\Pi_2} \underbrace{- \sum_{i=l+1}^{|\boldsymbol{y}|} \sum_{v \in [|\mathcal{V}|]} \boldsymbol{y}_{i,v} \log \pi_{LLM}(\boldsymbol{e}_v|\boldsymbol{y}_{\leq i-1}, \boldsymbol{x})}_{\Pi_3}, \tag{10}
$$

where $\frac{\partial \Phi_1}{\partial \boldsymbol{y}_l} = 0$ and $\frac{\partial \Pi_1}{\partial \boldsymbol{y}_l} = 0$ because they do not involve $\boldsymbol{y}_l$. $\Pi_2$ only depends on $\boldsymbol{y}_l$ through the term $\boldsymbol{y}_{l,v}$ while $\Pi_3$ only depends on $\boldsymbol{y}_l$ via $\log \pi_{LLM}(\boldsymbol{e}_v|\boldsymbol{y}_{\leq i-1}, \boldsymbol{x})$. Next, we compute the gradients for $\Phi_2$, $\Pi_2$, and $\Pi_3$, respectively.

$$
\boldsymbol{\delta}_{reward} = \frac{\partial \Phi_2}{\partial \boldsymbol{y}_l} = \sum_{i=l}^{|\boldsymbol{y}|} \frac{\partial r(\boldsymbol{y}_{\leq i}|\boldsymbol{x})}{\partial \boldsymbol{y}_l}, \tag{11}
$$

$$
\boldsymbol{\delta}_{prefix} = \frac{\partial \Pi_2}{\partial \boldsymbol{y}_l} = [-\log \pi_{LLM}(\boldsymbol{e}_1|\boldsymbol{y}_{\leq i-1}, \boldsymbol{x}), \cdots, -\log \pi_{LLM}(\boldsymbol{e}_N|\boldsymbol{y}_{\leq i-1}, \boldsymbol{x})]^{\top} \tag{12}
$$

$$
= -\log \mathrm{Cat}\left(\pi_{LLM}(\cdot|\boldsymbol{y}_{\leq l-1}, \boldsymbol{x})\right), \tag{13}
$$

$$
\boldsymbol{\delta}_{postfix} = \frac{\partial \Pi_3}{\partial \boldsymbol{y}_l} = -\sum_{i=l+1}^{|\boldsymbol{y}|} \sum_{v \in [|\mathcal{V}|]} \frac{\partial \log \pi_{LLM}(\boldsymbol{e}_v|\boldsymbol{y}_{\leq i-1}, \boldsymbol{x})}{\partial \boldsymbol{y}_l} \boldsymbol{y}_{l,v} \tag{14}
$$

$$
= -\sum_{i=l+1}^{|\boldsymbol{y}|} \frac{\partial \log \mathrm{Cat}\left(\pi_{LLM}(\cdot|\boldsymbol{y}_{\leq i-1}, \boldsymbol{x})\right)}{\partial \boldsymbol{y}_l} \boldsymbol{y}_l, \tag{15}
$$

as desired. □

*Remark* C.2. Our proposed DTO fundamentally differs from previous works that utilize gradients for controlled generation (Qin et al., 2022; Kumar et al., 2021; 2022; Mireshghallah et al., 2022; Liu et al., 2023b), where $\boldsymbol{\delta}_{postfix}$ is often detached from the computational graph, and only prior context is used to guide subsequent token prediction.

## C.2 PROOF OF THEOREM 4.1

**Theorem C.3** (Restatement of Theorem 4.1). *Suppose $\{\rho^t\}_{t\geq 0}$ denotes the Wasserstein gradient flow minimizing Eq. 3 in the distribution space with boundary conditions $\rho^0 = \pi_{LLM}$ and $\rho^\infty = \rho^* = \arg\min_\rho \mathcal{L}_{PPO}(\rho)$. Then we can draw samples from $\rho^*$ by first initializing $\boldsymbol{x}^0 \sim \pi_{LLM}$ and simulating a trajectory $\{\boldsymbol{x}^t\}_{t\geq 0}$ following the stochastic gradient flow of Eq. 2: $\frac{d\boldsymbol{x}^t}{dt} = -\nabla \mathcal{L}(\boldsymbol{x}^t) + \sqrt{2}\boldsymbol{\epsilon}_t$, where $\{\boldsymbol{\epsilon}_t \in \mathcal{N}(\boldsymbol{0}, \boldsymbol{I})\}_{t\geq 0}$ are Brownian motions.*

*Proof.* First of all, we derive the Wasserstein gradient flow for $\rho^t$ on $\mathbb{W}_2(\mathbb{R}^{L_x \times N})$ under the functional cost $\mathcal{L}_{PPO}$ (Chen et al., 2018):

$$
\partial_t \rho^t = -\nabla_{\mathbb{W}} \mathcal{L}_{PPO}(\rho^t) \quad \Rightarrow \quad \partial_t \rho^t + \nabla_{\boldsymbol{x}} \cdot \left(\rho^t \nabla_{\boldsymbol{x}} \frac{\delta \mathcal{L}_{PPO}}{\delta \rho^t}\right) = 0, \tag{16}
$$

where $\frac{\delta \mathcal{L}_{PPO}}{\delta \rho^t}$ denotes the first variation of $\mathcal{L}_{PPO}$ in terms of $\rho_t$, and the $\nabla \cdot$ is the divergence operator. We derive the first variation as below:

$$
\frac{\delta \mathcal{L}_{PPO}}{\delta \rho^t} = \frac{\delta}{\delta \rho^t} \left[\int -\lambda r(\boldsymbol{x}) + \log \frac{\rho^t(\boldsymbol{x})}{\pi_{LLM}(\boldsymbol{x})} \rho^t(\boldsymbol{x}) d\boldsymbol{x}\right] \tag{17}
$$

$$
= -\lambda r(\boldsymbol{x}) + \log \rho^t(\boldsymbol{x}) - \log \pi_{LLM}(\boldsymbol{x}) + 1. \tag{18}
$$

The gradient of $\frac{\delta \mathcal{L}_{PPO}}{\delta \rho^t}$ can be expressed as:

$$\nabla_{\boldsymbol{x}} \frac{\delta \mathcal{L}_{PPO}}{\delta \pi_t} = \nabla_{\boldsymbol{x}} \left( -\lambda r(\boldsymbol{x}) + \log \rho^t(\boldsymbol{x}) - \log \pi_{LLM}(\boldsymbol{x}) \right). \tag{19}$$

Now we substitute the above equations into Eq. 16 and find the following partial differential equation of $\rho^t$:

$$\partial_t \rho^t + \nabla_{\boldsymbol{x}} \cdot \left[ \rho^t \nabla_{\boldsymbol{x}} \left( -\lambda r(\boldsymbol{x}) + \log \rho^t(\boldsymbol{x}) - \log \pi_{LLM}(\boldsymbol{x}) \right) \right] = 0 \tag{20}$$

$$\partial_t \rho^t + \nabla_{\boldsymbol{x}} \cdot \left[ \rho^t \left( -\lambda \nabla_{\boldsymbol{x}} r(\boldsymbol{x}) - \nabla_{\boldsymbol{x}} \log \pi_{LLM}(\boldsymbol{x}) \right) \right] + \sum_{i,j=1}^{L_x, N} \frac{\partial}{\partial \boldsymbol{x}_{i,j}} \left( \nabla_{\boldsymbol{x}} \log \rho^t(\boldsymbol{x}) \right) = 0 \tag{21}$$

$$\partial_t \rho^t + \nabla_{\boldsymbol{x}} \cdot \left[ \rho^t \left( -\lambda \nabla_{\boldsymbol{x}} r(\boldsymbol{x}) - \nabla_{\boldsymbol{x}} \log \pi_{LLM}(\boldsymbol{x}) \right) \right] + \Delta \rho^t(\boldsymbol{x}) = 0, \tag{22}$$

where $\Delta$ means the Laplacian operator $\sum_{ij} \frac{\partial^2}{\partial \boldsymbol{x}_{ij}^2}$. By Fokker-Plank equation (Maoutsa et al., 2020), we obtain a velocity field for particles $\boldsymbol{x}^t \in \mathbb{R}^{L_x \times N}$:

$$\frac{d\boldsymbol{x}^t}{dt} = -\lambda \nabla_{\boldsymbol{x}} r(\boldsymbol{x}) - \nabla_{\boldsymbol{x}} \log \pi_{LLM}(\boldsymbol{x}) + \sqrt{2}\boldsymbol{\epsilon}_t = -\nabla_{\boldsymbol{x}} \mathcal{L}(\boldsymbol{x}^t) + \sqrt{2}\boldsymbol{\epsilon}_t, \tag{23}$$

where $\{\boldsymbol{\epsilon}_t\}_{t \geq 0}$ denotes a Brownian motion (Øksendal, 2003)).

Conversely, if $\boldsymbol{x}^t$ follows the above dynamics starting from the initial distribution $\boldsymbol{x}^t \sim \rho^0$, then the density function $\rho^t$ of $\boldsymbol{x}^t$ follows the time evolution in Eq. 16 (see Maoutsa et al. (2020)). Since we assume the limiting condition $\rho^t \to \rho^*$, we conclude that $\boldsymbol{x}^t \sim \rho^*$ in distribution as $t \to \infty$. □

### C.3 JUSTIFICATION OF CONFIDENCE-BASED TOKEN SELECTION

We re-parameterize the policy as a categorical distribution through a softmax function to ensure differentiability. In Sec. 3.3, we mention that we can skip optimizing tokens whose logits is over-confident as gradient descent is unlikely to significantly change its resultant distribution. We provide theoretical evidence for this argument.

Without loss of generality, we consider the scenario where we only optimize a single token $\boldsymbol{x} \in \mathbb{R}^{|\mathcal{V}|}$ within a vocabulary $\mathcal{V}$. We initialize a logit vector $\boldsymbol{z} \in \mathbb{R}^N$, then apply the softmax transformation to obtain the corresponding categorical distribution:

$$\boldsymbol{x}_i = \frac{\exp(\boldsymbol{z}_i)}{\sum_{j=1}^{|\mathcal{V}|} \exp(\boldsymbol{z}_j)}, \quad \forall i \in [|\mathcal{V}|]. \tag{24}$$

The loss function is defined over $\boldsymbol{x}$ and by chain rule, we derive the gradient w.r.t $\boldsymbol{z}_i$ for $i \in [|\mathcal{V}|]$:

$$\frac{\partial \mathcal{L}}{\partial \boldsymbol{z}_i} = \frac{\partial \mathcal{L}}{\partial \boldsymbol{x}} \frac{\partial \boldsymbol{x}}{\partial \boldsymbol{z}_i} = (\text{diag}(\boldsymbol{x}) - \boldsymbol{x}\boldsymbol{x}^\top)_i \frac{\partial \mathcal{L}}{\partial \boldsymbol{x}} = \boldsymbol{x}_i \left( \left[ \frac{\partial \mathcal{L}}{\partial \boldsymbol{x}} \right]_i - \boldsymbol{x}^\top \frac{\partial \mathcal{L}}{\partial \boldsymbol{x}} \right), \tag{25}$$

where we use the fact that the Jacobian matrix of softmax function is $\text{diag}(\boldsymbol{x}) - \boldsymbol{x}\boldsymbol{x}^\top$. The derivation above indicates that the gradient magnitude for the $i$-th logit is proportional to its corresponding post-softmax probability. When $\boldsymbol{x}_i$ is small at the initialization, then its underlying representation $\boldsymbol{z}_i$ cannot be updated effectively. This limitation underscores the necessity of skipping tuning $\boldsymbol{x}$ with high confidence throughout the decoding stage.

## D MORE ON EXPERIMENT

### D.1 EXPERIMENT DETAILS

We used a temperature of $0.5$ and a top-p of $0.95$, and set the maximum generation length to 1024 for AMC and MATH-500, and 3072 for AIME, for all baselines and our methods. We report the average performance across 4 independent runs on smaller datasets such as AMC, AIME-24, and AIME-25. For SFT experiments, we randomly sampled a 10k subset from the Open-thoughts dataset (Guha et al., 2025). For GRPO experiments, we adopted a random 35k subset from the Numina-Math

dataset (Li et al., 2024). Exclusively for $\nabla$-Reasoner, we set $\epsilon_{ent} = 0.25$, $\epsilon_{grad} = 8$, learning rate to 0.01, and number of iterations to 20 in all experiments. We use Skywork-Reward-V2-Qwen3-4B (Liu et al., 2025) as the reward model for Qwen family and Skywork-Reward-V2-Llama-3.1-8B (Liu et al., 2025) as the reward model for Llama model. Below, we list a more comprehensive summary of the experiment setups.

Table 4: Summary of Experimental Settings.

| Setting | Value |
|---|---|
| **Generation Hyperparameters** | |
| Temperature | 0.5 |
| Top-p | 0.95 |
| Max Generation Length (AMC, MATH-500) | 1024 |
| Max Generation Length (AIME) | 3072 |
| **Evaluation** | |
| Independent Runs (MATH-500) | 1 |
| Independent Runs (AMC, AIME-24, AIME-25) | 4 |
| **Training Data Subsets** | |
| SFT | 10k random sample from Open-thoughts  (Guha et al., 2025) |
| GRPO | 35k random sample from Numina math (Li et al., 2024) |
| **$\nabla$-Reasoner Hyperparameters** | |
| $\epsilon_{ent}$ | 0.25 |
| $\epsilon_{grad}$ | 8 |
| Learning Rate | 0.01 |
| Number of Iterations | 20 |
| Optimizer | Adam-W |
| LR Scheduler | Cosine |
| Min LR | 0.001 |
| **Reward Models** | |
| For Qwen Family | Skywork-Reward-V2-Qwen3-4B (Liu et al., 2025) |
| For Llama Model | Skywork-Reward-V2-Llama-3.1-8B (Liu et al., 2025) |

## D.2 Effectiveness of Acceleration Techniques

In Sec. 3.3, we proposed three major techniques to improve the decoding efficiency of $\nabla$-Reasoner. In this section, we quantify the contribution of each technique to the overall speedup. Instead of ablating them in running time (since removing any one of these components causes the algorithm to run in an prohibitive amount of time, often more than several hours per sample), we monitor how each technique reduces the cost of the specific stage to which it is applied. As a result, we find our gradient-caching mechanism bypasses more than 63.8% of the model calls in parallel forms required for gradient acquisition, and trajectory reuse eliminates more than 74.1% of autoregressive model calls. Both techniques exploit the sparsity of token updates. The token-selection strategy further complements these components by skipping the full optimization procedure when appropriate; we observe that it effectively avoids 89.2% of token-optimization steps across the sequence.

## D.3 Wall-clock Time Benchmarking under Idealized Settings

In this section, we measure and compare the wall-clock running time of our method with a strong baseline, BoN. We consider an idealized deployment setting where dynamic serving frameworks (e.g., vLLM) and batching are fully utilized, so that all prompts are processed in parallel. Under this assumption, both generation and optimization requests can be executed as batched operations. To estimate the running

Table 5: Comparison of wall-clock execution time for 83 prompts on the AMC dataset.

| Method | Gen. (s) | Optim. (s) | Total (s) | FLOPs |
|---|---|---|---|---|
| BoN ($N = 8$) | 136.1 | — | 136.1 | $9.54 \times 10^{15}$ |
| $\nabla$-**Reasoner** | 106.2 | 45.9 | 152.1 | $2.46 \times 10^{17}$ |

time in this setting, we first record execution traces of the reasoning process for both BoN and $\nabla$-Reasoner. The trace includes statistics such as the number of generated tokens per request, the number of gradient computations, and the number of tokens involved in optimization. We then group

and replay these traces to simulate parallel generation and optimization. The recorded statistics are used to reconstruct the execution process. The simulation sends generation requests to a vLLM server and performs gradient computation over batched sequences. We measure the corresponding wall-clock running time together with the computational FLOPs. The experiment was conducted on the AMC dataset using the Qwen-2.5-math-Instruct-7B model running on 8x 140GB NVIDIA H200 GPUs. The results are demonstrated in Tab. 5. As shown in our benchmarks, for a batch of 83 prompts, the BoN baseline ($N = 8$) requires 136.1 seconds of total execution time. In comparison, our approach completes the generation phase in 106.2 seconds, with an additional 45.9 seconds of overhead for the iterative optimization process (calculated as the total optimization time distributed across 8 GPUs). This results in a total wall-clock time of 152.1 seconds. We further analyze the computational efficiency by comparing the total FLOPs. While $\nabla$-Reasoner performs significantly more operations than the BoN baseline ($2.46 \times 10^{17}$ vs. $9.54 \times 10^{15}$ FLOPs), the total wall-clock time remains comparable. This is because hardware utilization (TFLOPS/s) during the DTO phase – where attention is executed in parallel – is significantly higher than during the auto-regressive phase, where attention must be computed sequentially.

### D.4 A Simplified Example on GSM8K

The problem asks:

> Josh buys a house for $80,000$ and then puts in $50,000$ in repairs. This increased the value of the house by $150\%$. How much profit did he make?

**Greedy Decoding Baseline (Failure Analysis)**

The original output reads:

```
Josh buys a house for $80,000 and then puts in $50,000 in repairs.
The value of the house increased by 150%, so the new value of the
house is 80,000 * 1.5 = $120,000. Josh sold the house for 120,000,
so he made a profit of 120,000 - 80,000 - 50,000 = -10,000. The
answer is -10,000.
```

This greedy decoding baseline fails by interpreting "increased by" as simple multiplication ($80,000 \times 1.5$), resulting in:
$$\text{Incorrect New Value} = \$80,000 \times 1.5 = \$120,000$$
and an incorrect profit of:
$$\text{Incorrect Profit} = \$120,000 - \$80,000 - \$50,000 = -\$10,000$$

**DTO-Based Token Revision (Correction)**

Our method corrects this error through DTO-based refinement.

Initial Input and Refinement Round 1 (Operator Correction)

- Initial Output Segment: "..., so the new value of the house is $80,000 \times 1.5 = \$120,000$."
- The algorithm performs refinement at the multiplication token "$\times$".
- Through gradient descent, the model updates the multiplication token "$\times$" to the addition symbol "$+$", revising the logical structure to: "..., so the new value of the house is $80,000 + \ldots$"

Refinement Round 2 (Calculation Correction)

- Subsequently, when calculating the increment, the model initially outputs $80,000 \times 1.5 = \$120,000$. The calculation is incorrect because the model ignores the preceding $80,000+$.
- The algorithm flags the token **120** (representing $\$120,000$) due to a low score.
- Gradient optimization identifies **200** as the top replacement, correcting the calculation for the new value:
  $$\text{Correct New Value} = \$80,000 + (\$80,000 \times 1.5) = \$80,000 + \$120,000 = \$200,000$$

Final Correct Derivation

The model completes the string to form "$200,000", allowing it to derive the correct profit:

$$\text{Correct Profit} = \$200,000 - \$80,000 - \$50,000 = \$70,000$$

**Conclusion**

In conclusion, after two rounds of iterative refinement, the final output becomes:

```
Josh buys a house for $80,000 and then puts in $50,000 in repairs.
The value of the house increased by 150%, so the new value of the
house is 80,000 + 80,000 * 1.5 = $200,000. Josh sold the house for
200,000, so he made a profit of 200,000 - 80,000 - 50,000 = 70,000.
The answer is 70,000.
```

