# OpenReview forum: "$\nabla$-Reasoner: LLM Reasoning via Test-Time Gradient Descent in Latent Space"
_ICLR.cc/2026/Conference — ICLR 2026 Poster_

### Official Review · Reviewer_qoEv · 2025-11-01

**Soundness:** 1
**Presentation:** 3
**Contribution:** 2
**Rating:** 4
**Confidence:** 4

**Summary:**

This paper introduces $\nabla$-Reasoner, a test-time differentiable reasoning framework for large language models (LLMs). It proposed Differentiable Textual Optimization (DTO) that performs gradient descent on logits. The paper provides some empirical experiments and theoretical analysis.

**Strengths:**

The idea sounds interesting. It is interesting to perform training on logits during test time since it has some challenges. The authors also proposed methods to reduce the heavy computational overhead, showing their awareness of practical utility.

**Weaknesses:**

My major concern is that **the experiment setting is considerably vague, incomplete and flawed** in many aspects.

1. **Lack of efficiency analysis**. This is a strong concern I have on this paper. Although the authors proposed methods to reduce the heavy overhead, the detailed analysis and experiments are missing. In Sec 5.1, “Efficiency” is measured solely by model calls, ignoring **wall-clock latency, FLOPs, and GPU memory**, which are the core components of efficiency. Intuitively it is very likely that the overhead would be significant since getting gradients would require at least 2x GPU memory, and the latency is likely substantial since it requires performing gradient descent back of forth. In addition, it is important to show whether the efficiency is strongly dependent on the choice of policy models and reward models, and the output sequence length. Lack of such analysis clearly undermines the soundness of the paper.

2. **Unfair comparison with test-time methods**. It is unfair to compare your methods with vanilla SC and BoN using both 8 rollouts. With tools like vLLM library, vanilla SC and BoN can be done flashly in one go, which is much more efficient than your proposed framework if the number of rollouts are the same. I suggest the authors align the budgets (wall-clock latency) with these methods to ensure fairness.

3. **Training-based baselines are not convincing**. In practice, we often apply SFT then RL, not to perform them separately. It's unfair to compare SFT and GRPO separately since they should be incorporated as a whole framework. The paper also did not demonstrate how the budget of the training-based baselines aligns with $\nabla$-reasoner.

4. **No comparison with other test-time training / inference-time adaptation baselines**. This is a serious issue since there are already so many test-time training methods, including but not limited to [1][2]. **None of these are compared with $\nabla$-reasoner.**

5. **Limited domain coverage**. All datasets are about math reasoning. It is unclear how the method performs on other domains like coding, general reasoning, open-ended QA, helpfulness and safety.

6. **Missing analysis of dependence on reward model quality**. The proposed method’s effectiveness appears highly sensitive to the quality and compatibility of the reward model (RM) used for gradient guidance. Appendix D shows that different RMs are applied for different policy LLMs. Why do you do that? Can your framework still work well when the quality of RM is not that good or the output of policy model is OOD for the RM? I suggest incorporating detailed experiments on the effect of applying different RMs across different tasks to show that whether your method is sensitive with the choice of RM or not.

7. **Missing details for reproducibility**. The paper incorporates many experimental details in Appendix D, which is good. However some details are still unclear, including hardware (which is crucial for efficiency analysis) and evaluation methods (rule-based evaluation or strict string matching?). The authors also did not provide their code.


In addition to the experiment settings, I also have concern on the theoretical analysis. Theorem 4.1 is not sound enough since there is a **clear gap between the theorem and actual methodology**. The Wasserstein gradient-flow theorem assumes continuous time, continuous space, infinitesimal steps, and Gaussian diffusion noise. However, actual DTO uses finite discrete steps and deterministic updates. Hence, the theory is **intuitive but not rigorous**, that could act as a motivational study but not a rigorous proof of the equivalence.

[1] Test-Time Preference Optimization: On-the-Fly Alignment via Iterative Textual Feedback  (ICML 2025)

[2] Learning to Reason from Feedback at Test-Time  (ACL 2025 main)

**Questions:**

1. What do you mean in "it reduces costs by up to 40.2%" in Sec 5.1's cost analysis? How did you get the figure? Which baseline are you comparing with? Is it about wall-clock latency or FLOPs or something else? This claim is not supported by any concrete experiments in the paper.

2. As mentioned above, why do you use different RMs for different policy LMs? Would the performance degrade when the quality of RM is poor or the output of policy model is OOD for the RM?

3. Is the overhead significant especially when RMs are large or output sequence is long?

4. Can your theory be more aligned with the actual methodology?

5. Could you provide some case studies, that is, concrete examples on what your framework is doing on some certain questions and models?

---

> ### Author Response · Authors · 2025-11-21
>
> Dear Reviewer qoEv,
>
> Thanks for carefully reviewing our work. Below are our responses to resolve your concerns.
>
> **1. Detailed efficiency analysis and fair comparison with baselines**
> We fully understand the reviewer's concern regarding efficiency evaluation. Efficiency has also been a challenge in our experiment evaluation. The difficulty lies in identifying a common and faithful metric (e.g., wall-clock time, FLOPs) that can compare our methods with baselines while still highlighting the actual potential of our approach under a prototypical implementation.
>
> The core dilemma is the following. A key advantage of $\nabla$-Reasoner is its ability to leverage both parallel and recurrent execution of transformers for decoding and for scaling test-time compute. The parallel execution naturally consumes an extensive FLOPs, yet often achieves small practical runtime due to improved hardware utilization (e.g. flash-attention). Consequently, our method can, in principle, exploit more compute under the same time budget compared with inference-only approaches.
>
> However, as we frankly admitted in the limitation section, our work focuses on algorithmic-level design, while system- and hardware-level optimizations, including integration with high-performance serving frameworks, fall outside the scope of this work. This gap means that simple inference-based methods already tightly optimized in existing LLM serving systems may achieve faster real-time performance even though their compute utility per time unit might be lower. We believe that future serving systems that support more flexible execution patterns for LLMs offer a promising path to further improving the efficiency of our approach.
>
> To address this, we use the number of model calls as a surrogate and theoretical efficiency metric. This choice is motivated by the observation that a single model call, whether executed in parallel or recurrent form, can have comparable wall-clock cost under ideal attention parallelization. Hence, it could be used as a unifying metric by mitigating discrepancy in system-level implementation. Under this assumption, the metric cleanly reflects the potential of $\nabla$-Reasoner to convert additional compute more effectively into improved performance. Throughout the paper, our efficiency claims are all based on this metric. We will involve this discussion and clarify our every claim on efficiency analysis.
>
> Guided by this metric, we further designed a series of techniques to improve efficiency (Sec. 3.3). $\nabla$-Reasoner substantially reduces recurrent model calls by branching the autoregressive generation at the correct intermediate position based on gradient-based signals, rather than always regenerating from the beginning. Our gradient caching techniques further decrease the number of required model evaluations.
>
> As a result, even when we measure actual wall-clock time, we observe that $\nabla$-Reasoner achieves on-par runtime with BoN when the same $N_{\max}$ is used, despite offering a strictly more expressive mechanism for scaling test-time compute.
>
> Table D1. End-to-end wall-clock time per sample (in seconds). The experiment was conducted on the AIME-25 dataset using the Qwen-2.5-Instruct-7B model running on eight 80GB NVIDIA A100 GPUs. We intentionally disable dynamic serving techniques (such as vLLM) across all generation steps in order to isolate and evaluate the effect of our method without the consideration of system-level optimizations.
> | Method | BoN ($N=8$) | SC ($N=8$) | Ours ($N_{\max}=8$) |
> |:-:|:-:|:-:|:-:|
> |Wall-clock Time | 40.3 s|  39.5 s | 41.1 s |
>
> While we set $N_{max} = 8$ to match the BoN baseline, we observe that most problems do not require all $N_{max}$ regeneration steps when solved by $\nabla$-Reasoner. Instead, $\nabla$-Reasoner uses gradient information to adaptively decide when to resample subsequence tokens and when to retain the current prediction.

---

> ### Author Response · Authors · 2025-11-21
>
> **2. Comparison with training-based methods.**
> We note that RL or GRPO is not necessarily applied only after SFT. In fact, appling RL alone has become a promising and increasingly popular setting for studying math reasoning capabilities (e.g., [1, 2]). Our goal here is also to isolate the contribution of each post-training techniques.
>
> The training cost and inference cost are fundamentally not comparable. Even though we intentionally choose moderate-sized datasets for training-time baselines, their total FLOPs and wall-clock time remain substantially larger than those of all inference-time methods. Our purpose of including training-time methods is to illustrate how improved inference-time techniques can approach the performance of training-based approaches at dramatically lower cost. To emphasize this point more clearly, we will include above discussion and add the estimated training FLOPs or time in our efficiency analysis.
>
> [1] Guo et al., Deepseek-r1: Incentivizing reasoning capability in llms via reinforcement learning.
>
> [2] Yue et al., Does Reinforcement Learning Really Incentivize Reasoning
> Capacity in LLMs Beyond the Base Model?
>
> **3. Comparison with self-refinement based methods**
>
> We thank the reviewer for suggesting additional baselines. The referenced approach differs fundamentally from ours: their "textual optimizatio" is only analogous in spirit, as the optimization loop is implemented via a self-refinement process of a pre-trained LLM, without using true gradient information. In contrast, our textual optimization explicitly leverages real gradient signals computed from the model.
>
> For comparison, we include the results of [1]. To roughly match the computational cost of BoN and our method with $N_{max} = 8$, we set the number of samples per step as $2$ and the number of refinement steps to $2$ for the approach in [1].
>
> Table D2: Performance comparison with Test-Time Preference Optimization (TPO). Experiments are conducted on AIME-2025 using the Qwen-2.5-Instruct-7B model.
> | Method | TPO ($N_{samples}=2$, $N_{refinement}=2$) | Ours ($N_{\max}$=8) |
> |:-:|:-:|:-:|
> |Accuracy| 11.1 | 20.0 |
>
>
> We will also mention the mentioned works in our related work.
>
> [1] Test-Time Preference Optimization: On-the-Fly Alignment via Iterative Textual Feedback
>
>
> **4. Performance on more domains**
> While our primary objective is to improve the mathematical reasoning capabilities of LLMs, we note that our approach also achieves competitive performance on coding tasks.
>
> We use `Qwen2.5-7B-Instruct` as the base policy and compare $\nabla$-Reasoner with BoN on the LeetCode benchmark [1]. The results are reported below:
>
> <!-- ==Table and results== -->
>
> Table D3: Performance comparison on the LeetCode dataset. Results show that our method consistently surpasses all baseline methods.
> | Method | LeetCode |
> |:-:|:-:|
> |Greedy decoding | 46.9% |
> |SC |48.2% |
> |BoN | 47.4% |
> |Ours | 50.0% |
>
>
> We will continue to expand our evaluation to include a more complete set of baselines and additional benchmarks to provide a comprehensive comparison.
>
> [1] Guo et al., DeepSeek-Coder: When the Large Language Model Meets Programming--The Rise of Code Intelligence
>
> **5. Dependence on reward model quality**
>
> We apply Skywork-Reward-V2-Qwen3-4B to models in the Qwen family and Skywork-Reward-V2-Llama-3.1-8B to models in the Llama family because our algorithm requires the reward model and the base model to share the same vocabulary. Otherwise, the gradient from the reward model would be incorrectly propagated to the logits produced by the base policy without an additional step aligning their vocabulary set.
>
> To study the dependency on reward-model quality, we further evaluate our approach on Qwen2.5-Math-7B-Instruct paired with the larger Skywork-Reward-V2-Qwen3-8B reward model. The results are shown below in Table D4.
>
>
> Table D4: Ablation study on reward model choice. Using MATH-500 as the validation set, we observe no significant performance differences across reward models.
> | Reward Model | `Skywork-Reward-V2-Qwen3-4B` |`Skywork-Reward-V2-Qwen3-8B` |
> |:-:|:-:|:-:|
> |ACC| 80.4 | 80.8 |
>
> We note that our original choice `Skywork-Reward-V2-Qwen3-4B` is a smaller and weaker reward model according to [RewardBench](https://huggingface.co/spaces/allenai/reward-bench). Based on the results above, we find that using a smaller reward model does not lead to significant performance degradation compared with the larger, stronger version. This justifies our original choice and further suggests that smaller reward models may even be preferable for improving efficiency.

---

> ### Author Response · Authors · 2025-11-21
>
> **6. Implementation details.**
>
> All our experiments are conducted on 8 × 80G A100 GPUs. We follow a standard evaluation pipeline to measure mathematical reasoning performance: we extract each model’s final answer as the string inside `\box{}` and compare it with the ground truth using the verifier from PRIME-RL. We measure Pass@1 accuracy averaged by multiple runs. We will release our code upon acceptance.
>
> **7. Is the overhead significant especially when RMs are large or output sequence is long?**
>
> Increasing the size of the reward model inevitably introduces extra cost for reward evaluation. However, as shown in our results, smaller reward models achieve comparable performance, suggesting that adopting a smaller RM is often a more desirable and economical choice. Increasing the sequence length likewise increases the cost per model call. Nevertheless, our method incorporates a strict selection mechanism to determine which tokens should be skipped or optimized, and we employ a gradient caching strategy that substantially reduces the number of model calls required during token optimization. Henceforth, in Table D5, we observe the optimzation time remain constant when handling long sequence outputs as the generation time grow linearly.
>
> Table D5: Total time breakdown for short vs. relatively long samples from the AIME-2025 dataset. "Generation time" denotes the total time required to produce new tokens, while "optimization time" refers to the time spent optimizing tokens via DTO. The results show that optimization time remains stable as generation length increases, whereas generation time grows approximately linearly.
> | Generation Length | Generation Time | Optimization Time |
> |:-:|:-:|:-:|
> | 1368 | 54.8 s | 26.9 s |
> | 2128 | 82.7 s | 28.1 s |
>
>
> **8. Gap between theory and practical implementation.**
> The purpose our Theorem 4.1 is to provide insights on how test-time gradient descent in textual space connects to RL training algorithms. It is not intended as a convergence guarantee or a formal theoretical bound for our approach. Rather, it aims to illuminate how training-time RL and test-time textual optimization can be understood within a unified conceptual framework.
> In particular, our DTO algorithm can be viewed as a discretized version of the gradient flow derived from the Wasserstein gradient flow on policy. It is common in the literature to study complex training dynamics by analyzing more tractable SDEs or ODEs [1]. Moreover, prior work has established a clean correspondence between certain SDEs and their noise-free ODE counterparts (e.g. [2]). In our setting, the optimization space remains continuous because gradients are backpropagated to the token logits.
>
> [1] Song et al., A mean field view of the landscape of two-layer neural networks
> [2] Song et al., Score-based generative modeling through stochastic differential equations
>
>
> **9. Concrete examples.**
>
> Due to space limit, we provide one simplified example in GSM8k dataset as below:
>
> The problem asks:
> ```
> Josh buys a house for $80,000 and then puts in $50,000 in repairs. This increased the value of the house by 150%. How much profit did he make?
> ```
> The original output reads:
> ```
> Josh buys a house for $80,000 and then puts in $50,000 in repairs. The value of the house increased by 150%, so the new value of the house is 80,000 * 1.5 = $120,000. Josh sold the house for 120,000, so he made a profit of 120,000 - 80,000 - 50,000 = -10,000. The answer is -10,000.
> ```
>
> This greedy decoding baseline fails by interpreting "increased by" as simple multiplication (80,000 * 1.5), resulting in a house value of $120,000 and an incorrect profit of -$10,000.
>
> Our method corrects this error through an iterative token revision process:
> - It takes the intial intput: *"..., so the new value of the house is 80,000 * 1.5 = $120,000."*
> - The algorithm performs refinement at the multiplication token `*`.
> - Through gradient descent, the model updates the multiplication token `*` to the addition symbol `+`, revising the logical structure to "..., so the new value of the house is 80,000 +".
> - Subsequently, when calculating the increment, the model initially outputs "80,000 * 1.5 = $120,000". The calculation is incorrect because the model ignores the preceding `80,000 +`.
> - The algorithm flags the token `120` due to a low score, and the gradient optimization identifies `200` as the top replacement, correcting the calculation to "80,000 * 1.5 = $200".
> - Finally, the model completes the string to form "$200,000", allowing it to derive the correct profit: *"$200,000 - $80,000 - $50,000 = $70,000."*
>
> In conclusion, after two rounds of iterative refinement, the final output becomes:
> ```
> Josh buys a house for $80,000 and then puts in $50,000 in repairs. The value of the house increased by 150%, so the new value of the house is 80,000 + 80,000 * 1.5 = $200,000. Josh sold the house for 200,000, so he made a profit of 200,000 - 80,000 - 50,000 = 70,000. The answer is 70,000."
> ```

---

> ### Comment · Reviewer_qoEv · 2025-11-22
> **Thanks but unfortunately I am still not convinced enough at this stage**
>
> Thank you to the authors for the detailed and timely responses. I appreciate the additional experiments and clarifications provided. However, after carefully reviewing the rebuttal and the newly included results, I unfortunately believe that my **fundamental concerns remain unresolved**, and therefore my assessment of this submission **stays below the acceptance bar**. I summarize the key issues below.
>
> ### **1. Efficiency analysis and system-level considerations remain insufficient**
>
> I understand that this work focuses on an algorithmic contribution rather than a systems paper.
> However, $\nabla$-Reasoner introduces substantially more complexity than BoN or SC, and this complexity directly impacts its practical deployability.
>
> Since BoN and SC already benefit from **highly mature and widely adopted system optimizations (e.g., vLLM)**, their end-to-end latency profile in practical deployments differs dramatically from the abstract “model call” metric used in this paper. These are **not** “special optimizations,” but the **standard runtime environments** used across the community.
>
> I therefore believe the paper should at minimum report BoN/SC latency **under standard vLLM optimization**, and explicitly acknowledge that $\nabla$-Reasoner is currently incompatible with these optimizations, thereby limiting its real-world usability.
>
> Additionally, my concerns about **memory usage, FLOPs analysis, and GPU footprint**, which are all crucial for a method relying heavily on test-time backpropagation, were not addressed. These metrics are essential for evaluating such a decoding-heavy approach.
>
> ### **2. The argument for excluding SFT+RL baselines remains unconvincing**
>
> Multiple recent works (e.g., [3][4]) demonstrate the importance of SFT+RL combinations. **Even the works cited by the authors do not justify the claim that combining SFT and RL is unnecessary**:
>
> * [1] clearly shows that **lack of an SFT cold start leads to instability and poor readability** (DeepSeek-R1-Zero), and DeepSeek-R1 ultimately **does** use SFT as a cold start.
> * [2] only argues that SFT may not be mandatory for math tasks in that specific experimental design (but not arguing that excluding SFT is beneficial; only to isolate the effect of RL for study). It explicitly states SFT is essential for domains such as code or multimodal reasoning.
>
> Moreover, The proposed method **fails to outperform GRPO on roughly half the datasets. Only **Qwen-2.5-7B** is compared with GRPO.** Why are **Qwen-2.5-7B-Instruct** and **Llama-3.1-8B-Instruct** excluded from this comparison?
>
> Without these baselines, the claim of competitive or superior performance is not well supported.
>
> [1] Guo et al., Deepseek-r1: Incentivizing reasoning capability in llms via reinforcement learning.
>
> [2] Yue et al., Does Reinforcement Learning Really Incentivize Reasoning Capacity in LLMs Beyond the Base Model?
>
> [3] Chu et al., SFT Memorizes, RL Generalizes: A Comparative Study of Foundation Model Post-training
>
> [4] Jin et al., RL Fine-Tuning Heals OOD Forgetting in SFT
>
> ### **3. “Not using true gradients” is not a valid reason to exclude strong baselines**
>
> The rebuttal suggests certain methods should not be compared because they do not use “true gradient information.” However, **not requiring gradient computation is often an advantage**, not a limitation since it induces lower computational cost, smaller memory footprint, and stronger compatibility with modern inference systems. These are areas where ∇-Reasoner is currently weaker, so excluding such baselines is not justified.
>
> ### **4. Coding-domain results remain too weak and the baselines insufficient**
>
> Thank you for including coding-domain experiments. However, **the gains over greedy decoding are small, and the comparison is limited only to SC and BoN (toy baselines).** Stronger baselines (RAP, ToT, Training ...) are necessary to support your claims.
>
> ### **5. Reward-model diversity is limited, and vocabulary matching restricts flexibility**
>
> All reward models tested are from the **Skywork-Reward-V2-Qwen** family and differ only in scale.
> This does not demonstrate robustness across reward modeling paradigms. Important families such as Math-Shepherd, Eurus, RLHFlow were not evaluated. If strict vocabulary matching is required to obtain usable gradients, this introduces another significant limitation: poor flexibility across tasks and model families, and needs to train a new reward model from scratch for every new backbone, which increases engineering and computational cost. This limitation is not acknowledged in the paper.
>
> As a small suggestion: ICLR allows authors to upload a **revised PDF during the rebuttal period**, with up to one additional page. Given the number of missing experiments and clarifications, I believe the authors may want to consider submitting a revised PDF to consolidate all new evidence and acknowledge the limitations more explicitly.

---

### Official Review · Reviewer_wy2L · 2025-11-01

**Soundness:** 3
**Presentation:** 3
**Contribution:** 3
**Rating:** 6
**Confidence:** 3

**Summary:**

This paper investigates expanding computation during inference. Existing methods tend to saturate in long-chain reasoning due to the influence of sparse and noisy rewards. The authors propose performing gradient descent on token logits in the sample space during decoding. Theoretically, DTO supports bidirectional gradient propagation along the sequence and establishes an equivalence to reinforcement learning–based training. Empirically, it outperforms baselines across multiple benchmarks, achieving comparable or superior reasoning accuracy at a lower computational cost.

**Strengths:**

1.The method demonstrates outstanding performance across multiple benchmarks.
2.The authors made practical considerations, enabling the proposed method to integrate well with existing LLM inference acceleration infrastructures.
3.The authors derived gradients over discrete text and attempted to provide theoretical guarantees for DTO

**Weaknesses:**

1. The authors introduced a hyperparameter in the objective function but did not conduct experiments to analyze its impact.
2.The proposed method heavily relies on the performance of the reward model, yet only one reward model was used in the experiments. We hope to see results under multiple reward models.

**Questions:**

See weakness

---

> ### Author Response · Authors · 2025-11-21
>
> Dear Reviewer wy2L,
>
> Thanks for your effort on reviewing our paper. We are grateful that you highlight both of our theoretical ideas and practical considerations. Below are our responses to further resolve your concern.
>
> **1. Impact of hyperparameters $\lambda$.**
> $\lambda$ is the hyperparameter in the DTO algorithm that balances the strength of the reward signal against the likelihood constraint imposed by the base model. As shown in Theorem 4.1, $\lambda$ also directly controls the strength of the KL regularization term: smaller $\lambda$ values keep the optimization closer to the original policy, while larger values allow more aggressive updates. We validate this behavior empirically in the Table C1.
>
> Table C1: Ablation study on hyperparameter choices. We report accuracy on the MATH-500 dataset using the Qwen-2.5-Math-Instruct model.
> |$\lambda$| 10 | 100 (default choice) | 1000 |
> |:-:|:-:|:-:|:-:|
> | ACC | 71.4\% | 80.4\% | 71.2\% |
>
> The results confirm that overly large values of $\lambda$ push the optimization toward sequences that falls out of the language model’s distribution and the policy updated by DTO will be all rejected, while overly small values limit the strength of the refinement signal. Our chosen setting lies near a sweet spot.
>
> **2. Results on multiple reward models.**
> We additional consider using a stronger reward model Skywork-Reward-V2-Qwen3-8B as suggested by RewardBench. We combine it with Qwen2.5-Math-7B-Instruct and the comparison are shown in Table C2. While the stronger reward model offers a slight improvement, the gain remains marginal. Overall, our method is robust across different reward-model choices, and given the efficiency benefits of smaller models, $\nabla$-Reasoner in practice tends to favor smaller and faster reward models.
>
> Table C2: Ablation study on reward model choice. Using MATH-500 as the validation set, we observe no significant performance differences across reward models.
> | Reward Model | `Skywork-Reward-V2-Qwen3-4B` |`Skywork-Reward-V2-Qwen3-8B` |
> |:-:|:-:|:-:|
> |ACC| 80.4 | 80.8 |

---

### Official Review · Reviewer_gxvJ · 2025-11-01

**Soundness:** 3
**Presentation:** 3
**Contribution:** 3
**Rating:** 6
**Confidence:** 3

**Summary:**

This paper focues on test-time scaling to improve LLM's performance for reasoning task. Commonly, we can use a reward model to choose the best sampling from a pool of candidates (BoN). Such a method can be considered as zeroth-order search. In this paper, the authors propose a new iterative generation framework, ∇-Reasoner, that applies differentiable optimization over token logits into the decoding loop to redfine the policy with the signal from reward model. The Differentiable Textual Optimization (DTO) can be considered as a first-order optimization, used at test time. It utilizes the gradient signals from the LLM's likelyhood and reward model to refine the textual representations.

The proposed method is theoretically justified, and empirically shows significantly better accuracy and efficiency than various strong baselines.

**Strengths:**

1. The paper is well-written, with a clear motivation.
2. The proposed method, DTO, is interesting and novel, offering a new way based on gradient optimization to test-time scaling. And DTO is supported with theoretical justification.
3. Extensive experiments show a significant accuracy and efficiency improvement.

**Weaknesses:**

1. Lack of reward models. In Appendix D, I notice you apply different reward models for differetn policy models. May I ask why? Could you offer an ablation study for the choice of different reward models for the same policy model.
2. The improvement statement in the abstract is overstated.  The 20% accuracy improvement and 40% less computation are for different baselines, incuring some confusion.

**Questions:**

See weakness

---

> ### Author Response · Authors · 2025-11-21
>
> Dear Reviewer gxvJ,
>
> Thank you for reviewing our paper and admitting our work is interesting and novel. Below is our responses to further resolve your concern.
>
> **1. The choice of reward models.**
> We apply Skywork-Reward-V2-Qwen3-4B to models in the Qwen family and Skywork-Reward-V2-Llama-3.1-8B to models in the Llama family because our algorithm requires the reward model and the base policy to share the same vocabulary. Otherwise, the gradient produced by the reward model would be misaligned with the logits of the base model, unless an additional vocabulary-alignment step were introduced.
> We offer additional results on applying a stronger reward model Skywork-Reward-V2-Qwen3-8B (according to RewardBench) to Qwen2.5-Math-7B-Instruct.
> We report the result in Table B1. We observe that although a stronger reward model yields a modest improvement, the gain remains marginal. Our approach is robust to the choice of reward model, and given the efficiency advantages of smaller models, $\nabla$-Reasoner in practice even prefers smaller and faster reward models.
>
> Table B1: Ablation study on reward model choice. Using MATH-500 as the validation set, we observe no significant performance differences across reward models.
> | Reward Model | `Skywork-Reward-V2-Qwen3-4B` |`Skywork-Reward-V2-Qwen3-8B` |
> |:-:|:-:|:-:|
> |ACC| 80.4 | 80.8 |
>
> **2. Overstated improvement.**
> We thank the reviewer for raising this concern. We will revise our abstract and introduction carefully to make a more moderate summary.

---

### Official Review · Reviewer_sgwy · 2025-11-02

**Soundness:** 3
**Presentation:** 3
**Contribution:** 3
**Rating:** 6
**Confidence:** 3

**Summary:**

This paper introduces ∇-Reasoner, an inference-time reasoning framework that applies gradient-based optimization to refine LLM outputs during decoding. The core innovation is Differentiable Textual Optimization (DTO), which optimizes token logits via gradient descent using signals from both the LLM's likelihood and a reward model. The approach combines an objective that balances reward maximization with log-likelihood regularization (Eq. 2), employs the straight-through estimator for differentiability, and integrates rejection sampling to accept only improved tokens. To enhance efficiency, the authors introduce gradient caching, rollout trajectory reusing, and confidence/gradient-guided token selection. Theoretically, they establish that sampling from an RL-optimized LLM is equivalent to drawing from the reference LLM and refining via DTO's gradient flow (Theorem 4.1). Evaluated on mathematical reasoning benchmarks (MATH-500, AMC, AIME), ∇-Reasoner achieves over 20% accuracy improvement while reducing computation by approximately 40% compared to Best-of-N and Self-Consistency baselines, demonstrating superior test-time scaling efficiency.

**Strengths:**

Originality:
- The shift from zeroth-order search to first-order gradient-based optimization for test-time reasoning is conceptually appealing and well-motivated by Figure 1
- The theoretical connection between DTO and PPO via Wasserstein gradient flow (Theorem 4.1) provides an elegant unification of parametric and non-parametric inference perspectives
- The gradient decomposition (δ_prefix, δ_postfix, δ_reward) offers clear intuition about how DTO enables bidirectional information flow along sequences

Quality:
- The experimental evaluation spans multiple model families (Qwen-2.5, Llama-3.1) and benchmarks with consistent improvements
- The ablation study in Table 2 effectively demonstrates DTO's impact on rejection rates (reducing from 66% to ~30-40%)
- The cost-accuracy trade-off analysis (Figure 3, Figure 4) convincingly shows efficiency gains over sampling-based baselines
- The comparison includes both test-time methods (SC, BoN, ToT, RAP) and training-based methods (SFT, GRPO), providing comprehensive context

Clarity:
- The motivation is clearly articulated—gradient information provides richer directional guidance than scalar rewards
- Figure 2 and Algorithms 1-2 effectively communicate the overall framework
- The progression from basic formulation to acceleration techniques is logical and well-structured
- The gradient decomposition in Section 4 provides valuable interpretability

Significance:
- Addresses the fundamental limitation of sparse reward signals in test-time scaling
- The 40% cost reduction while maintaining or improving performance has practical value
- The theoretical framework connecting test-time and training-time optimization could inspire future work
- The method appears general enough to work with different reward models and base LLMs

**Weaknesses:**

Experimental Analysis:
- While the cost comparison uses "number of calls," actual wall-clock time could differ due to backward passes. Could you provide runtime measurements to complement the theoretical cost analysis?
- The comparison with RAP and ToT might not be entirely fair if those methods weren't given comparable computational budgets. Could you ensure all baselines use similar total compute?
- Table 1 shows ∇-Reasoner sometimes underperforms training-based GRPO (e.g., Qwen-2.5-7B on AMC: 51.5% vs 52.8%). This suggests potential limitations—could you characterize when training-time methods remain superior?

Implementation Details:
- The method requires modifying the decoding loop and maintaining gradients through generation, which could complicate integration into existing serving infrastructure. Have you explored compatibility with standard inference optimization (e.g., KV caching, speculative decoding)?
- The acceleration techniques are crucial for practicality but add complexity. It would be valuable to quantify each technique's individual contribution to overall speedup
- For the Llama-3.1 experiments, you note inability to evaluate on AIME due to model incapability. This suggests potential brittleness—how does the method degrade with weaker base models?

Comparison Fairness:
- The SFT baseline uses only 10K examples while GRPO uses 35K—this asymmetry makes it difficult to assess whether ∇-Reasoner truly matches training-based methods or is being compared against undertrained baselines
- For the "comparable performance with training-based methods" claim, some results show ∇-Reasoner trailing GRPO (e.g., AIME25 on Qwen-2.5-7B: 15.0% vs 16.7%)

**Questions:**

- Gradient quality with straight-through estimator: The straight-through trick provides biased gradients. Have you investigated whether this bias significantly impacts convergence, or explored alternative differentiable relaxations (e.g., Gumbel-softmax)?
- Reward model dependency: How does performance degrade with reward models of varying quality? Have you experimented with different reward model architectures or training strategies? Could process rewards provide better gradients than outcome rewards?
- Optimization landscape: Can you characterize the reward landscape's properties (smoothness, multimodality, local optima)? Does DTO sometimes get stuck in local optima, and if so, how might this be addressed?

---

> ### Author Response · Authors · 2025-11-21
>
> Dear Reviewer sgwy,
>
> We appreciate your time on reviewing our work and the acknowledgement of our originality, quality, writing clarity, and significance. We are also grateful for your constructive comments. Please see our responses below:
>
> **1. Runtime efficiency measurements**
> To report runtime measurements and compare against existing baselines, we provide the table below showing the average wall-clock time on the AIME dataset.
>
> Table A1: End-to-end wall-clock time per sample (in seconds). The experiment was conducted on the AIME-25 dataset using the Qwen-2.5-math-Instruct-7B model running on eight 80GB NVIDIA A100 GPUs. We intentionally disable dynamic serving techniques (such as vLLM) across all generation steps in order to isolate and evaluate the effect of our method without the consideration of system-level optimizations.
> | Method | BoN ($N=8$) | SC ($N=8$) | Ours ($N_{\max}=8$) |
> |:-:|:-:|:-:|:-:|
> |Wall-clock Time | 40.3 s|  39.5 s | 41.1 s |
>
> Even though our approach currently demonstrates a similar running time to BoN, we emphasize that $\nabla$-Reasoner has greater potential to leverage compute/FLOPs more effectively by executing transformers in parallel mode. While BoN can benefit from highly optimized serving engines, we speculate that integrating the optimization procedure of $\nabla$-Reasoner better with the generation pipeline would create a much larger efficiency gap between our method and inference-only approaches.
>
> **2. Computational budgets of RAP and ToT.**
> We control the computational budget of BoN, SC-CoT, and our method by limiting the maximum number of sampled sequences, ensuring they operate under comparable compute. In contrast, the computational budget for RAT and ToT is more difficult to estimate and therefore harder to strictly match. To ensure strong performance and capability, we directly adopt the hyperparameters adopted in [1] for these methods. As shown in our runtime measurements below, RAP and ToT in fact exhibit significantly higher running time than our approach.
>
> Table A2: End-to-end wall-clock time per sample (in seconds) comparison between our method and RAT and ToT. Similar to previous experiment, the times are measured on the AIME-25 dataset using the Qwen-2.5-math-Instruct-7B model running on eight 80GB NVIDIA A100 GPUs. We intentionally disable dynamic serving techniques (such as vLLM) across all generation steps in order to isolate and evaluate the effect of our method without the consideration of system-level optimizations.
> | Method | RAP | ToT | Ours ($N_{\max}=8$) |
> |:-:|:-:|:-:|:-:|
> |Wall-clock Time | 264.7 s | 150.8 s | 41.1 s |
>
> [1] Hao et al., LLM Reasoners: New Evaluation, Library, and Analysis of Step-by-Step Reasoning with Large Language Models
>
> **3. When training-time methods remain superior?**
> In our experiments, training-based methods require substantially more computation to achieve the reported performance, whereas our method enhances test-time scaling techniques to reach a comparable performance. The performance difference relative to GRPO is insignificant, especially considering the additional compute GRPO demands. However, we emphasize that $\nabla$-Reasoner (like many test-time techniques) is complementary to training-based methods. As our Theorem 4.1 suggests, policy tuning through SFT or GRPO improves a model’s overall performance across the entire data distribution, whereas our approach operates on a per-sample basis. Training-time methods amortize their performance gains over the full population, while test-time optimization is particularly effective for further boosting performance on challenging individual problems. We will include this discussion in our revision.
>
> **4. Compatibility with standard LLM serving infrastructure**
>
> We acknowledge that our method is not directly compatible with existing LLM serving infrastructures such as vLLM. However, many of their key features are already leveraged by $\nabla$-Reasoner. For example, our decoding pipeline alternates between autoregressive decoding and gradient-based optimization. The autoregressive component can fully benefit from prefix caching provided by such serving engines, while the gradient-based optimization can be initialized using the logits obtained from them. The primary challenge, however, is that optimization and generation currently have to operate as two separate systems in our implementation. A tighter integration supporting batching and in-memory sharing would further improve throughput and reduce communication overhead. As we note in the limitations section, developing such an integrated serving solution remains an important direction for future exploration.

---

> ### Author Response · Authors · 2025-11-21
>
> (cont.)
>
> **5. Quantify each techniques' contribution to overall speedup.**
>
> We thank the reviewer for suggesting this interesting comparison. In practice, we find that the overall speedup arises from the synergy of all three techniques, and removing any one of these components causes the algorithm to run in an prohibitive amount of time, often more than several hours per sample. Instead of ablating them in overall running time, we monitor how each technique reduces the cost of the specific stage to which it is applied. Our gradient-caching mechanism bypasses more than $63.8\\%$ of model calls required for gradient acquisition, and trajectory reuse eliminates more than $74.1\\%$ of autoregressive model calls. Both techniques exploit the sparsity of token updates. The token-selection strategy further complements these components by skipping the full optimization procedure when appropriate; we observe that it effectively avoids $89.2\\%$ of token-optimization steps across the sequence.
>
> **6. Whether the compared training baselines are under-trained?**
> We note that 10k samples for SFT and 35k samples for GRPO are often regarded as a standard setting and sufficient to make meaningful improvement over the base policy as demonstrated by [1,2,3]. In fact, we intentionally choose moderate-sized datasets to control the cost spent for post-training baselines. Even though, their training cost is far more than inference-time methods, our approach reaches on-par accuracy with these methods.
>
> [1] Muennighoff et al., s1: Simple test-time scaling
> [2] Guha et al., OpenThoughts: Data Recipes for Reasoning Models
> [3] Luo et al., Deepscaler: Surpassing o1-preview with a 1.5 b model by scaling RL
>
>
> **7. Other differentiation relaxation other than straight-through estimator.**
>
> We cite [1] as one of the earliest works introducing surrogate backward computation for non-differentiable operations. However, the actual mechanism we use, described in Lines 217–219, is in fact called the hard Gumbel–Softmax relaxation. We adopt the "hard" variant because our gradient caching technique relies on the fact that the output of this operation changes only when the argmax switches. As a result, the gradient back-propagated from the model remains unchanged for most steps, allowing us to significantly reduce the number of required model calls. We will cite [2] in addition to improve clarity.
>
> [1] Bengio et al., Estimating or propagating gradients through stochastic neurons for conditional computation
> [2] Jang et al., Categorical Reparameterization with Gumbel-Softmax
>
>
> **8. Base model and reward model dependency**
>
> $\nabla$-Reasoner, like all existing inference-time techniques, can only effectively gain performance by scaling compute when the base model already possesses preliminary capability on the target task. Interestingly, as shown in Table 1, $\nabla$-Reasoner can substantially boost the performance of a weaker base model (Llama-3.1-8B), enabling it to significantly surpass a stronger model (Qwen-2.5-Math-7B). In contrast, post-training approaches such as SFT are often more limited in closing this gap.
> In contrast, our new experiments (shown in the table A4) indicate that using a stronger reward model does not yield notable accuracy improvements. This suggests that employing a relatively smaller reward model is preferable, as it provides better efficiency without sacrificing performance.
>
> Table A3: Ablation study on reward model choice. Using MATH-500 as the validation set, we observe no significant performance differences across reward models.
> | Reward Model | `Skywork-Reward-V2-Qwen3-4B` |`Skywork-Reward-V2-Qwen3-8B` |
> |:-:|:-:|:-:|
> |ACC| 80.4 | 80.8 |
>
> **9. Optimization landscape.**
>
> Theoretically characterizing the global optimization landscape is extremely challenging due to the high dimensionality and complexity of language models. However, we observe that our algorithm effectively leverages local properties of the reward landscape to optimize tokens. Full-sequence updates via DTO are inherently difficult, but making localized adjustments that better align tokens with the reward signal is far more stable and consistently guides the base policy toward a locally high-reward region. Based on these empirical findings, we view a more systematic characterization of the reward landscape (in the spirit of [1]) as an important direction for future work.
>
> [1] Razin et al., What Makes a Reward Model a Good Teacher? An Optimization Perspective

---

### Author Response · Authors · 2025-12-03

We sincerely thank all reviewers for their time and effort in providing valuable feedback on our work. We are honored that our submission was recognized as clearly written (Reviewers sgwy, gxvJ), well-motivated (Reviewers sgwy, gxvJ), technically novel (Reviewers sgwy, gxvJ, wy2L, qoEv), theoretically sound (Reviewers sgwy, wy2L), and empirically significant (Reviewers sgwy, gxvJ, wy2L). During the rebuttal, we addressed the major concerns raised and revised the manuscript accordingly (all changes marked in blue). Specifically, the major updates are:

1. **Wall-clock time efficiency.**
Our original efficiency claims were based on the number of model calls per task, a metric that serves as a theoretical surrogate capturing both parallel and recurrent execution of transformers. To address concerns from Reviewers sgwy and qoEv, we have (i) refined our claims to be precise with respect to this metric, (ii) clarified its definition and motivation, explaining why it is appropriate for demonstrating the potential of our approach to leverage compute more effectively, and (iii) added new experiments measuring real wall-clock running time for our method and baselines (see Tab. 2).

2. **Dependence on reward-model quality.**
To address concerns from Reviewers sgwy, gxvJ, wy2L, and qoEv regarding sensitivity to reward-model quality, we evaluated our approach using a stronger reward model (Tab. 3). The results show that our method performs reasonably well even with a smaller and potentially weaker reward model, supporting the default use of the smaller model for improved efficiency.

3. **Experimental details and additional comparisons.**
We have expanded the experimental section with additional implementation details, including hyperparameter settings for both baselines and our models. Moreover, we have included TPO as an additional iterative-refinement baseline for more comprehensive comparison.

However, we respectfully disagree with Reviewer qoEv’s assessment that incompatibility with current serving engines (e.g., vLLM) constitutes a major weakness and a reason for rejection. We believe that LLM infrastructures should ultimately adapt to serve inference algorithms -- rather than constraining algorithmic innovation. Although our work focuses on the algorithmic side, we have included detailed techniques to minimize overhead (Sec. 3.3). At the same time, we note that our algorithms are in fact amenable to existing infrastructures. For example, the iterative decoding process is fully compatible with current serving engines, and the DTO procedure can leverage transformer parallelism very effectively. The primary remaining gap is that optimization and decoding pipelines currently exist as separate system components. Bridging this gap requires additional engineering effort to fuse these components so that batching and resource reuse (e.g., KV caching) are supported seamlessly. Our algorithm can in fact further enable more dynamic on-device compute exploitation once such integration is developed.

Thanks for the time and effort invested in reading our responses!

Best,

Authors

---

### Meta-Review · Area_Chair_aY1G · 2026-01-02

**Summary:**

This paper proposes $\nabla$-Reasoner, a test-time reasoning framework that replaces discrete sampling-based search with first-order optimization in token/logit space. The core idea is to perform gradient-based refinement of generated reasoning traces using a reward model, combined with rejection sampling and several efficiency-oriented heuristics. The paper provides a theoretical connection between this procedure and KL-regularized RL, and presents empirical results on mathematical reasoning benchmarks showing consistent improvements over strong test-time baselines, with claimed gains in compute efficiency under a model calls metric. Across reviews, there was agreement that the core idea is novel and interesting, the paper is clearly written, and the experimental results demonstrate real performance gains on challenging reasoning tasks. The main points of contention concerned the interpretation of efficiency claims, compatibility with existing inference systems, and the scope and fairness of baseline comparisons.

Overall, this paper presents a genuinely novel approach to scaling LLM reasoning at test time, with a strong theoretical framing and convincing empirical gains on standard reasoning benchmarks. While there are legitimate concerns about practical efficiency, system integration, and the scope of comparisons, these are best viewed as limitations rather than fundamental flaws, in my opinion. A poster acceptance is appropriate because it acknowledges the strength and originality of the contribution, while recognizing that further validation around real-world efficiency and deployment will be important for broader adoption.

**Reviewer Concerns:**

Concerns largely addressed by rebuttal:
- Questions about sensitivity to reward model quality were addressed with additional ablations using stronger reward models, showing only marginal differences in performance.
- Requests for clarification on hyperparameters and additional experimental details were handled well.
- The authors added wall-clock runtime measurements and clarified the intent and limitations of the number of model calls efficiency metric.
- Some concerns about missing comparisons (e.g. additional iterative refinement baselines) were partially addressed by adding results for TPO and expanded discussion.

Concerns partially or not fully resolved:
- A key unresolved concern is the practical efficiency and deployability of the approach. While the authors are transparent about focusing on algorithmic design rather than systems integration, the method currently does not integrate cleanly with widely used inference frameworks, and the efficiency advantages rely on idealized assumptions about parallelism.
- There remains some disagreement about whether baseline comparisons particularly against training-based methods and other test-time adaptation approaches fully support the paper's broader claims.
- One reviewer remained unconvinced after rebuttal, primarily due to these system-level and evaluation-scope concerns.

**Reviewer Scores:**

- For sgwy the rebuttal addressed most technical questions, but core concerns about efficiency framing remain.
- For gxvJ, the rebuttal would likely make their score unchanged and stay at a 6.
- For wy2L the rebuttal would likely make their score unchanged and stay at a 6.
- qoEv already expressed they are not convinced so they would keep their 4.

---

### Decision · Program_Chairs · 2026-01-26

Accept (Poster)